# A Unifying Framework to the Analysis of Interaction Methods using Synergy Functions

## Abstract

Deep learning is expected to revolutionize many sciences and particularly healthcare and medicine. However, deep neural networks are generally "black box," which limits their applicability to mission-critical applications in health. Explaining such models would improve transparency and trust in AI-powered decision making and is necessary for understanding other practical needs such as robustness and fairness. A popular means of enhancing model transparency is to quantify how individual inputs contribute to model outputs (called attributions) and the magnitude of interactions between groups of inputs. A growing number of these methods import concepts and results from game theory to produce attributions and interactions. This work presents a unifying framework for game-theory-inspired attribution and $k^{\text{th}}$-order interaction methods. We show that, given modest assumptions, a unique full account of interactions between features, called synergies, is possible in the continuous input setting. We identify how various methods are characterized by their policy of distributing synergies. We establish that gradient-based methods are characterized by their actions on monomials, a type of synergy function, and introduce unique gradient-based methods. We show that the combination of various criteria uniquely defines the attribution/interaction methods. Thus, the community needs to identify goals and contexts when developing and employing attribution and interaction methods. Finally, experiments with Physicochemical Properties of Protein Tertiary Structure data indicate that the proposed method has favorable performance against the state-of-the-art approach.

[1]Anonymous Institution, Anonymous City, Anonymous Region, Anonymous Country. Correspondence to: Anonymous Author <anon.email@domain.com>.

Preliminary work. Under review by the International Conference on Machine Learning (ICML). Do not distribute.

## 1. Introduction

Explainability has become an ever increasing topic of interest among the Machine Learning (ML) community. Various ML methods, including deep neural networks, have unprecedented accuracy and functionality, but their models are generally considered "black box" and unexplained. Without "explaining" a model's workings, it can be difficult to troubleshoot issues, improve performance, guarantee accuracy, or ensure other performance criteria such as fairness.

A variety of approaches have been employed to address the explainability issue of neural networks. Taking the taxonomy of (Linardatos et al., 2020), some methods are universal in application (called model agnostic) (Ribeiro et al., 2016), while other are limited to specific types of models (model specific) (Binder et al., 2016). Some model-specific methods are limited to a certain data type, such as image (Selvaraju et al., 2017) or tabular data (Ustun & Rudin, 2016). Some methods are global, i.e., they seek to explain a model's workings as a whole (Ibrahim et al., 2019), while others are local, explaining how a model works for a specific input (Zeiler & Fergus, 2014). Finally, some methods seek to make models that are intrinsically explainable (Letham et al., 2015), while others, called post hoc, are designed to be applied to a black box model without explaining it (Springenberg et al., 2014). These post hoc methods may seek to ensure fairness, test model sensitivity, or indicate which features are important to a model's prediction.

This paper focuses on the concept of attributions and interactions. *Attributions* are local, post hoc explainbility methods that indicate which features of an input contributed to a model's output (Lundberg & Lee, 2017), (Sundararajan et al., 2017), (Sundararajan & Najmi, 2020), (Binder et al., 2016), (Shrikumar et al., 2017). *Interactions*, on the other hand, are methods that indicate which groups of features may have interacted, producing effects beyond the sum of their parts (Masoomi et al., 2021), (Chen & Ye, 2022), (Sundararajan et al., 2020), (Janizek et al., 2021), (Tsai et al., 2022), (Blücher et al., 2022), (Zhang et al., 2021), (Liu et al., 2020), (Tsang et al., 2020a), (Hamilton et al., 2021), (Tsang et al., 2020b), (Hao et al., 2021), (Tsang et al., 2017), (Tsang et al., 2018). A common and fruitful approach to attributions and interactions is to translate and apply results from

game theoretic cost sharing (Shapley & Shubik, 1971), (Aumann & Shapley, 1974). This has the advantages of already having a well-developed theory and producing methods that uniquely satisfy identified desirable qualities.

This work utilizes a game theoretic viewpoint to analyze, unify, and extend existing attribution and interaction methods. The contributions of this paper are as follows:

- This paper offers a method of analysis for attribution and $k^{\text{th}}$ order interaction methods of continuous-input models through the concept of synergy functions. We show that, given natural and modest assumptions, synergy functions give a unique accounting of all interactions between features. We also show any continuous input function has a unique synergy decomposition.
- We highlight how various (existing) methods are governed by rules of synergy distribution, and common axioms constrain the distribution of synergies. With this in mind, we highlight the particular strengths and weaknesses of established methods.
- We show that under natural continuity criteria, gradient-based attribution/interaction methods on analytic functions are uniquely characterized by their actions on monomials. This collapses the question "how should we define interactions on analytic functions" to "how should we define interactions of a monomial?" We then give two methods that serve as potential answers to this question.
- We discuss the goal-dependent nature of attribution and interaction methods. Based on this observation, we identify a method for producing new attributions and interactions.

## 2. Background

### 2.1. Notation and Terminology

Let $N = \{1, ..., n\}$ denote the set of feature indices in a machine learning model (e.g. pixel indices in an image classification model). For $a, b \in \mathbb{R}^n$, let $[a, b] = \{x \in \mathbb{R}^n : a_i \leq x_i \leq b_i \text{ for all } i \in N\}$ denote the hyper-rectangle with opposite vertices $a$ and $b$. Let $F : [a, b] \mapsto \mathbb{R}$ denote a machine learning model taking an input data point $x \in [a, b]$ and outputting a real number. For example, $F(x)$ can be viewed as the output of a softmax layer (for a specific class) in a neural network classifier. We denote the class of such functions by $\mathcal{F}(a, b)$, or $\mathcal{F}$ if $a, b$ may be inferred. Define a *baseline attribution method* as:

**Definition 1** (Baseline Attribution Method). A baseline attribution method is any function of the form $A(x, x', F) : D \to \mathbb{R}^n$, where $D \subseteq [a, b] \times [a, b] \times \mathcal{F}$. [1]

Baseline attribution methods give the contribution of each feature in an input feature vector, denoted $x \in [a, b]$, to a function's output, $F(x)$, with respect to some baseline feature vector $x' \in [a, b]$.[2] We denote a general baseline attribution by A, so that $A_i(x, x', F)$ is the attribution score

---

[1] Some attribution and interaction methods also incorporate the internal structure of a model. We do not consider these here.

of feature $x_i$ to $F(x)$, with respect to the baseline feature values $x'$. The definition allows for attributions with more restricted domains than $[a, b] \times [a, b] \times \mathcal{F}$ because baseline attributions may require conditions on $F$ or $x$ in order to be well defined. We will see a simple example of such conditions when we define Integrated Gradient method in section 2.3. For the purpose of this paper, all attribution methods are baseline attribution methods.

While attribution methods give a score to the contribution of each input feature, *Interactions* give a score to a group of features based on the group's contribution to $F(x)$ beyond the contributions of each feature (Grabisch & Roubens, 1999). For ease of reference, we may speak of a nonempty set $S \subseteq N$ as being a group of features, by which we mean the group of features with indices in $S$. Let $\mathcal{P}_k = \{S \subseteq N : |S| \leq k\}$ contain all subsets of $N$ of size $\leq k$. Then we can define a $k^{\text{th}}$-order baseline interaction method by:

**Definition 2** ($k^{\text{th}}$-Order Baseline Interaction Method). A $k^{\text{th}}$-order baseline attribution method is any function of the form $I^k(x, x', F) : D \to \mathbb{R}^{|\mathcal{P}_k|}$, where $D \subseteq [a, b] \times [a, b] \times \mathcal{F}$.

$k^{\text{th}}$-order interaction methods are a sort of expansion of attributions, giving a contribution for each group of features in $\mathcal{P}_k$. For some $S \in \mathcal{P}_k$, the term $I_S^k(x, x', F)$ indicates the component of $I^k(x, x', F)$ that gives interactions among the group of features $S$. When speaking of interactions among a group of features, there are multiple possible meanings: marginal interactions between members of a group, total interactions among members of the group, and average interactions among members of the group. Loosely speaking, if we let $G_S$ be the interactions among the features of $S$ that are not accounted for by the interactions of sub-groups, then $G_S$ represents marginal interactions of features in $S$, $\sum_{T \subseteq S} G_T$ represents the total interactions of features in $S$, and $\sum_{T \subseteq S} \mu_T G_T$ represents average interactions of features in $S$, where $\mu_T$ is some weight function. This paper focuses on marginal interactions.

Using quadratic regression as an example, suppose $F(x_1, x_2, x_3) = 2x_1 - 3x_2 + x_1 x_3 - 15$, $x = (1, 1, 1)$, $x' = (0, 0, 0)$. Then a $2^{\text{nd}}$-order baseline interaction method may report something like: $I_\emptyset(x, x', F) = -15$, $I_{\{1\}}(x, x', F) = 2$, $I_{\{2\}}(x, x', F) = -3$, and $I_{\{1,3\}}(x, x', F) = 1$, and the other interactions equal zero.

It should be noted that $1^{\text{st}}$-order interactions with $I_\emptyset^1$ disregarded and baseline attributions have equivalent definitions. As with attributions, interactions may not be defined for

---

[2] As an example, the first proposed baseline for image inputs was a black image, which corresponds to the zero vector (Sundararajan et al., 2017). The question of an appropriate baseline generally depends on the data. See Pascal Sturmfels (2020) for a survey of baselines for image tasks.

all $(x, x', F)$. We denote the set of inputs where a given $\mathrm{I}^k$ is defined by $D_{\mathrm{I}^k}$, or $D_{\mathrm{A}}$ with regard to attributions. As with attributions, all interactions are baseline $k^{\text{th}}$-order interactions for the purpose of this paper. We may drop $x'$ if the baseline is fixed, and also drop $x$, implying that some appropriate value is considered.

## 2.2. Axioms

The definitions provided in the previous subsection are extremely general and may lead to attribution functions that are not practical. To find practically-relevant attributions or interaction methods, the standard strategy is to identify certain axioms a method should satisfy. In this subsection, we review the common axioms of attributions and interactions used in prior work (Grabisch & Roubens, 1999) (Sundararajan et al., 2020), (Sundararajan & Najmi, 2020), (Tsai et al., 2022), (Janizek et al., 2021), (Marichal & Roubens, 1999), (Zhang et al., 2020). Axioms are only presented for interactions; they can be easily reformulated for attributions by setting $k = 1$ and disregarding $\mathrm{I}_{\emptyset}^1$, so that $\mathrm{I}^1(x, x', F) : D \to \mathbb{R}^n$.

1. **Completeness**: $\sum_{S \in \mathcal{P}_k, |S| > 0} \mathrm{I}_S^k(x, x', F) = F(x) - F(x')$ for all $(x, x', F) \in D_{\mathrm{I}^k}$.

*Completeness* is sometimes called efficiency in the game-theoretic literature and derives from the concept of cost-sharing (Shapley & Shubik, 1971),(Sundararajan et al., 2017). In attributions and interactions, requiring completeness grounds the meaning of the interaction values by requiring the method account for the total function value change $F(x) - F(x')$.

2. **Linearity**: If $(x, x', F), (x, x', G) \in D_{\mathrm{I}^k}$, $a, b \in \mathbb{R}$, then $(x, x', aF + bG) \in D_{\mathrm{I}^k}$, and $\mathrm{I}^k(x, x', aF + bG) = a\mathrm{I}^k(x, x', F) + b\mathrm{I}^k(x, x', G)$.

*Linearity* ensures that when a model is a linear combination of sub-models, the interactions or attributions of the model is a weighted sum of the interactions or attributions of the sub-models.

We say that a function $F \in \mathcal{F}$ does not vary in some feature $x_i$ if for any vector $x \in [a, b]$, $f(t) = F(x_1, .., x_{i-1}, t, x_{i+1}, ..., x_n)$ is constant. This indicates that $F$ is not a function of $x_i$. On the contrary, if it is false to say that $F$ does not vary in $x_i$, then we say $F$ varies in $x_i$. If $F$ does not vary in $x_i$, we call $x_i$ a null feature of $F$.

3. **Null Feature**: If $(x, x', F) \in D_{\mathrm{I}^k}$, $F$ does not vary in $x_i$, and $i \in S$, then $\mathrm{I}_S^k(x, x', F) = 0$.

*Null Feature* asserts that there is no marginal interaction among a group if one of the features has no effect. There may be interactions between subsets of $S$ so long as they do not contain a null feature.[3]

---

[3]Null feature is similar to dummy as stated in Sundararajan et al. (2017) and Sundararajan et al. (2020).

The three axioms above, completeness, linearity, and null features, are generally assumed in the literature on game-theoretic attributions and interactions. Besides these three, there are many other axioms (guiding principles) offered that generally serve one of two purposes: either they distinguish a method as unique, or they show that a method satisfies desirable qualities. Among them are symmetry (Sundararajan et al., 2020), symmetry-preservation (Sundararajan et al., 2017), (Janizek et al., 2021), (Sundararajan & Najmi, 2020), interaction symmetry (Janizek et al., 2021), (Tsai et al., 2022), interaction distribution (Sundararajan et al., 2020),(Sundararajan et al., 2020), sensitivity (sometimes called sensitivity (a))(Sundararajan et al., 2017), (Sikdar et al., 2021), implementation invariance (Sundararajan et al., 2017), (Sundararajan et al., 2020), (Janizek et al., 2021), (Sikdar et al., 2021), non-decreasing positivity (Lundstrom et al., 2022), recursive axioms (Grabisch & Roubens, 1999), (Tsai et al., 2022), faithfulness (Tsai et al., 2022), affine scale invariance (Friedman, 2004), (Sundararajan & Najmi, 2020), (Xu et al., 2020), demand monotonicity (Sundararajan & Najmi, 2020). Some of the above axioms, such as linearity or implementation invariance, are satisfied by many methods, but no one method satisfies all axioms. For example, Faith-Shap (Tsai et al., 2022) is characterized by a faithfulness criteria, while Shapley-Taylor (Sundararajan et al., 2020) is characterized by interaction distribution.

## 2.3. Attribution and Interaction Methods

Here we review several well known attribution and interaction methods based on cost sharing. Before we introduce them, we first introduce a necessary notation. For given features $S \subseteq N$ and assumed baseline $x'$, we define $x_S \in [a, b]$ by:

$$(x_S)_i = \begin{cases} x_i & \text{if } i \in S \\ x_i' & \text{if } i \notin S, \end{cases} \quad (1)$$

where $x_i$ is the $i^{\text{th}}$ element of $x$ and $x_i'$ is the $i^{\text{th}}$ element of $x'$. One well known attribution method is the **Shapley Value** (Shapley & Shubik, 1971), (Lundberg & Lee, 2017):

$$\text{Shap}_i(x, F) = \frac{1}{n} \sum_{S \subseteq N \setminus \{i\}} \binom{n-1}{|S|}^{-1} (F(x_{S \cup \{i\}}) - F(x_S)),$$

where $\binom{n-1}{|S|} \triangleq \frac{(n-1)!}{(n-1-|S|)!(|S|)!}$ denotes the number of subsets of size $|S|$ of $n - 1$ features. The Shapley value is an import of the famous Shapley value from game-theory in ML attributions. It is an example of a *binary features method*, meaning it only considers $F$ evaluated at the points $\{x_S : S \subseteq N\}$; that is, points where each feature value is the input or baseline value. Multiple $k^{\text{th}}$-order interactions that extend Shapley values have been proposed, all of which are binary feature methods (Grabisch & Roubens, 1999),(Tsai et al., 2022), (Sundararajan et al., 2020).

Another well known attribution is the **Integrated Gradients** (IG) (Sundararajan et al., 2017):

$$\text{IG}_i(x, F) = (x_i - x_i') \int_0^1 \frac{\partial F}{\partial x_i}(x' + t(x - x'))dt. \quad (2)$$

The IG is a direct translation of the well known cost-sharing method of Aumann-Shapley (Aumann & Shapley, 1974) to ML attributions. For the theoretical foundations of IG, see Sundararajan et al. (2017), Aumann & Shapley (1974), Lundstrom et al. (2022).

Currently, no $k^{\text{th}}$-order interactions extension of the IG has been proposed. However, a 2-order interaction, **Integrated Hessian** (IH), has been proposed in Janizek et al. (2021). This interaction method computes the pairwise interaction between $x_i$ and $x_j$ as:

$$\text{IH}_{\{i,j\}}(x, F) = 2(x_i - x_i')(x_j - x_j')$$
$$\times \int_0^1 \int_0^1 st \frac{\partial^2 F}{\partial x_i \partial x_j}(x' + st(x - x'))dsdt$$

The "main effect" of $x_i$, or lone interaction (a misnomer), is defined as:

$$\text{IH}_{\{i\}}(x, F) = (x_i - x_i') \times \int_0^1 \int_0^1 \frac{\partial F}{\partial x_i}(x' + st(x - x'))dsdt$$
$$+ (x_i - x_i')^2 \times \int_0^1 \int_0^1 st \frac{\partial^2 F}{\partial x_i^2}(x' + st(x - x'))dsdt$$

IH is what we label a *recursive method* since it uses an attribution method recursively. Specifically, $\text{IH}_{\{i,j\}}(x, F) = \text{IG}_i(x, \text{IG}_j(\cdot, F)) + \text{IG}_j(x, \text{IG}_i(\cdot, F))$. Similarly, $\text{IH}_{\{i\}}(x, F) = \text{IG}_i(x, \text{IG}_i(\cdot, F))$ (Janizek et al., 2021).

### 2.4. The Möbius Transform

Lastly, we review the Möbius transform, which will be useful for our definition of the notion of "pure interactions" in section 3. Let $v$ be a real-valued function on $|N|$ binary variables, so that $v : \{0,1\}^N \to \mathbb{R}$. For $S \subseteq N$, we write $v(S)$ to denote $v((\mathbb{1}_{1 \in S}, ..., \mathbb{1}_{n \in S}))$, where $\mathbb{1}$ is the indicator function. Recall that the Möbius transform of $v$ is a function $a(v) : \{0,1\}^N \to \mathbb{R}$ given by Rota (1964):

$$a(v)(S) = \sum_{T \subseteq S} (-1)^{|S| - |T|} v(T). \tag{3}$$

The Möbius transform satisfies the following relation to $v$:

$$v(S) = \sum_{T \subseteq N} a(v)(T)\mathbb{1}_{T \subseteq S} = \sum_{T \subseteq S} a(v)(T). \tag{4}$$

The Möbius transform can be conceptualized as a decomposition of $v$ into the marginal effects on $v$ for each subset of $N$. Each subset of $S$ has its own marginal effect on the change in function value of $v$, so that $v(S)$ is a sum of the individual effects, represented by $a(v)(T)$ in Eq. (4). For example, if $N = \{1, 2\}$, then for

$$v(S) = \begin{cases} \alpha & \text{if } S = \emptyset \\ \beta & \text{if } S = \{1\} \\ \gamma & \text{if } S = \{2\} \\ \delta & \text{if } S = \{1, 2\} \end{cases}$$

we have

$$a(v)(S) = \begin{cases} \alpha & \text{if } S = \emptyset \\ \beta - \alpha & \text{if } S = \{1\} \\ \gamma - \alpha & \text{if } S = \{2\} \\ \delta - \beta - \gamma + \alpha & \text{if } S = \{1, 2\} \end{cases}$$

## 3. Möbius Transforms as a Complete Account of Interactions

### 3.1. Motivation: Pure Interactions

In order to identify desirable qualities of an interaction method, it would be fruitful to answer the question: what sorts of function is a "pure interaction" of features in $S$? Specifically, is $F(x_1, x_2, x_3) = x_1 x_2$ a function of pure interaction between $x_1$ and $x_2$? This question is useful because if $F$ is a pure interaction of $x_1$ and $x_2$ (i.e. the only effects in $F$ is an interaction between $x_1$ and $x_2$), then naturally it ought to be that $\text{I}_S^2(x, F) = 0$ for $S \neq \{1, 2\}$. Indeed, to continue the example, suppose $F$ is a general function and we can decompose $F$ as follows:

$$F(x) = f_\emptyset + \sum_{1 \leq i \leq 3} f_{\{i\}}(x_i) + \sum_{1 \leq i < j \leq 3} f_{\{i,j\}}(x_i, x_j) + f_{\{1,2,3\}}(x),$$

where $f_\emptyset$ is some constant, $f_{\{i\}}$ is pure main effect of $x_i$; $f_{\{i,j\}}$ gives pure pairwise interactions; and $f_{\{1,2,3\}}$ is pure interaction between $x_1$, $x_2$, and $x_3$. Assuming $\text{I}^2$ conforms to linearity, we would gain:

$$\text{I}_S^2(x, F) = \sum_{|T| \leq 3} \text{I}_S^2(x, f_T) = \text{I}_S^2(x, f_S) + \text{I}_S^2(x, f_{\{1,2,3\}}),$$

by applying the above principle, namely $\text{I}_S^2(x, f_T) = 0$ if $S \neq T, |T| \leq 2$. That is, the $2^{\text{nd}}$-order interaction of $F$ for $S$ would be a sum of $\text{I}_S^2$ acting on the pure interaction function for group $S$, written $f_S$, and $\text{I}_S^2$ acting on a pure interaction of size 3. This would generalize to higher order interactions, so that:

$$\text{I}_S^k(x, F) = \text{I}_S^k(x, f_S) + \sum_{T \subseteq N, |T| > k} \text{I}_S^k(x, f_T).$$

We would then have to determine what rules should govern $\text{I}_S^k(x, f_S)$, and $\text{I}_S^k(x, f_T), |T| > k$.

### 3.2. Unique Full-Order Interactions

In the previous section we spoke intuitively regarding the notion of pure interaction; we now present a formal treatment. Let $\text{I}^n$ be a $n^{\text{th}}$-ordered interaction function, i.e., $\text{I}^n$ gives the interaction between all possible subsets of features. In addition to the axioms of completeness and null features above, we propose two modest axioms for such a function; first, we propose a milder form of linearity, which requires linearity only for functions that $\text{I}_S^n$ assign no interaction to. We weaken linearity in the interest of establishing the notion of pure interactions with minimal assumptions.

4. **Linearity of Zero-Valued Functions**: If $(x, x', G)$, $(x, x', F) \in D_{\text{I}^n}$, $S \subseteq N$ such that $\text{I}_S^n(x, x', G) = 0$, then $\text{I}_S^n(x, x', F + G) = \text{I}_S^n(x, x', F)$.

Before introducing the next axiom, we consider the meaning of the baseline, $x'$. In cost sharing, the baseline is the state where all agents make no demands (Shapley & Shubik, 1971). If an agent makes no demands, there are no attributions, nor are there interactions with other players. Likewise, the original IG paper notes (Sundararajan et al., 2017):

> *"Let us briefly examine the need for the baseline in the definition of the attribution problem. A common way for*

*humans to perform attribution relies on counterfactual intuition. When we assign blame to a certain cause we implicitly consider the absence of the cause as a baseline for comparing outcomes. In a deep network, we model the absence using a single baseline input."*

As with the cost sharing literature and Sundararajan et al. (2017), we interpret the condition $x_i = x_i'$ to indicate that the feature $x_i$ is not present. Now, for given features $S \subseteq N$ and assumed baseline $x'$, we define $x_S \in [a, b]$ by:

$$(x_S)_i = \begin{cases} x_i & \text{if } i \in S \\ x_i' & \text{if } i \notin S, \end{cases} \tag{5}$$

where $x_i$ is the $i^{\text{th}}$ element of $x$ and $x_i'$ is the $i^{\text{th}}$ element of $x'$. With this in mind, we present the next axiom:

5. **Baseline Test for Interactions** ($k = n$): For baseline $x'$, if $F(x_S)$ is constant $\forall x$, then $\mathrm{I}_S^n(x, x', F) = 0$.

This axiom states that if every variable $\notin S$ is held at the baseline value, and the other variables $\in S$ are allowed to vary, but the function is a constant, then there is no interaction between the features of $S$. Why is this sensible? The critical observation is that a feature being at its baseline value indicates the feature is not present. If the features of $S$ have no effect when other features are absent, then the features of $F$ do not interact in and of themselves and their interaction measurement should be zero.

Our setup allows $F$ and $x'$ to be chosen separately. However, it is generally the case that data and task will inform an appropriate choice of baseline. We proceed assuming that $x'$ is chosen as the fitting baseline to $F$.

We now present an key result in our analysis:

**Theorem 1.** There is a unique $n$-order interaction method with domain $[a, b] \times [a, b] \times \mathcal{F}$ that satisfies completeness, null feature, linearity of zero-valued functions, and baseline test for interactions ($n = k$).

Proof of Theorem 1 is deferred to Appendix D.1. We turn to explicitly defining the unique interaction function satisfying the conditions in Theorem 1. For a fixed $x$ and implicit $x'$, $F(x_S)$ is a function of $S$. This implies it can be formulated as a function of binary variables indicating whether each input component of $F$ takes value $x_i$ or $x_i'$. Thus we can take the Möbius transform of $F(x_{(\cdot)})$, written as $a(F(x_{(\cdot)}))$. Now, if we evaluate the Möbius transform of $F(x_{(\cdot)})$ for some $S$, given as $a(F(x_{(\cdot)}))(S)$, and allow $x$ to vary, then this is a function of $x$. Recall that $\mathcal{P}_k = \{S \subset N : |S| \le k\}$. Given a baseline $x'$, define the **synergy function**:

**Definition 3** (Synergy Function). For $F \in \mathcal{F}$, $S \in \mathcal{P}_n$, and implicit baseline $x' \in [a, b]$, the synergy function $\phi : \mathcal{P}_N \times \mathcal{F} \to \mathcal{F}$ is defined by the relation $\phi_S(F)(x) = a(F(x_{(\cdot)}))(S)$.

We present the following example to help illustrate the synergy function: let $F(x_1, x_2) = a + bx_1^2 + c \sin x_2 + dx_1 x_2^2$,

and suppose $x' = (0, 0)$ are the baseline values for $x_1$ and $x_2$ that indicate the features are not present. The synergy for the empty set is the constant $F(x') = a$, indicating the baseline value of the function when no features are present. To obtain $\phi_{\{1\}}(F)$, we allow $x_1$ to vary but keep $x_2$ at the baseline, and subtract the value of $F(x')$. This gives us $\phi_{\{1\}}(F)(x) = a + bx_1^2 - a = bx_1^2$. If instead we allow only $x_2$ to vary, we get $\phi_{\{2\}}(F)(x) = a + c\sin(x_2) - a = c\sin(x_2)$. Finally, if we allow both to vary and subtract of all the lower synergies, we get $\phi_{\{1,2\}}(F)(x) = dx_1 x_2^2$. With the above definition, we turn to the following corollary:

**Corollary 1.** The synergy function is the unique $n$-order interaction method that satisfies completeness, null feature, linearity of zero-valued functions, and baseline test for interactions ($n = k$).

Commentary on precursors to the synergy function and a proof of Corollary 1 are relegated to Appendices D.2 and D.3, respectively.

### 3.3. Properties of the Synergy Function

Given a function $F$, the synergy of a single feature $x_i$ is given by $\phi_{\{i\}}(F)(x) = F(x_{\{i\}}) - F(x')$, and the pairwise synergy for features $x_i$ and $x_j$ is

$$\phi_{\{i,j\}}(F)(x) = F(x_{\{i,j\}}) - \phi_{\{i\}}(F)(x) - \phi_{\{j\}}(F)(x) - F(x')$$
$$= F(x_{\{i,j\}}) - F(x_{\{i\}}) - F(x_{\{j\}}) + F(x').$$

In general, the synergy function for a group of features $S$ is

$$\phi_S(F)(x) = F(x_S) - \sum_{T \subsetneq S, T \ne \emptyset} \phi_T(F)(x) - F(x')$$
$$= \sum_{T \subseteq S} (-1)^{|S| - |T|} \times F(x_T)$$

With this we can define the notion of a pure interaction. A *pure interaction function* of the features $S$ is a function that 1) takes a value of 0 if any feature in $S$ takes its baseline value, and 2) varies and only varies in the features in $S$.[4] This is exactly what the synergy function accomplishes: either $\phi_S(F)(x) = 0$, or $\phi_S(F)(x)$ varies in exactly the features in $S$ and is 0 whenever $x_i = x_i'$ for any $i \in S$. More technically, define $C_S = \{F \in \mathcal{F} | F$ is a pure interaction function of $S\}$ to be the set of pure interactions of features $S$. Then we have the following corollary:

**Corollary 2.** Suppose an implicit baseline $x' \in [a, b]$ and let $F \in \mathcal{F}$, and $S, T \in \mathcal{P}_n$. Then the following hold:
1. Pure interaction sets are disjoint, meaning $C_S \cap C_T = \emptyset$ whenever $S \ne T$.
2. $\phi_S$ projects $\mathcal{F}$ onto $C_S \cup \{0\}$. That is, $\phi_S(F) \in C_S \cup \{0\}$ and $\phi_S(\phi_S(F)) = \phi_S(F)$.
3. For $\Phi_T \in C_T$, we have $\phi_S(\Phi_T) = 0$ whenever $S \ne T$.
4. $\phi$ uniquely decomposes $F \in \mathcal{F}$ into a set of pure interaction functions on distinct groups of features. That is,

---

[4]For the degenerate case where $S = \emptyset$, a pure interaction of the features of $S$ would be a constant function.

there exists $\mathcal{P} \subset \mathcal{P}_n$ such that $F = \sum_{S \in \mathcal{P}} \Phi_S$ where each $\Phi_S \in C_S$, only one such representation exists, and $\Phi_S = \phi_S(F)$ for each $S \in \mathcal{P}$ while $\phi_S(F) = 0$ for each $S \in \mathcal{P}_n \setminus \mathcal{P}$.

Proof of Corollary 2 is relegated to Appendix D.4. For ease of notation, we move forward assuming that if $x'$ is not stated, the implicit baseline value is $x' = 0$ and is appropriate to $F$. We also assume that the synergy functions $S$ is applied using the proper implicit baseline choice. Lastly, we denote $\Phi_S \in C_S$ to be a pure interaction in $S$ as defined above, or what we may also call a "synergy function" in $S$.

## 4. Binary Feature Methods and Synergies

We now discuss the role of the synergy function in axiomatic attributions/interactions. Harsanyi (1963)[5] noticed that for a synergy function $\Phi_S$, the Shapley value is

$$\text{Shap}_i(x, \Phi_S) = \begin{cases} \frac{\Phi_S(x)}{|S|} & \text{if } i \in S \\ 0 & \text{if } i \notin S \end{cases} \quad (6)$$

This means the Shapley value distributes the function gain from $\Phi_S$ equally among all $i \in S$. Using the synergy representation of $F$ and linearity of Shapley values, we get

$$\text{Shap}_i(x, F) = \sum_{S \subseteq N \, s.t. \, i \in S} \frac{\Phi_S(x)}{|S|} \quad (7)$$

Thus, the Shapley value can be conceptualized as distributing each synergy $\Phi_{\{i\}}$ to $x_i$ and distributing all higher synergies, $\Phi_S$ with $|S| \geq 2$, equally among all features in $S$, e.g., $\text{Shap}(\Phi_{\{1,2,3\}}) = (\frac{\Phi_{\{1,2,3\}}}{3}, \frac{\Phi_{\{1,2,3\}}}{3}, \frac{\Phi_{\{1,2,3\}}}{3}, 0, ..., 0)$. Indeed the Shapley value is characterized by its rule of distributing the synergy function.

**Proposition 1.** *(Grabisch, 1997, Thm 1)* The Shapley value is the unique attribution that satisfies linearity and acts on synergy functions as in (6).

Other binary-feature methods are similar. We present a treatment of Shapley-Taylor in Appendix E.1. We also present a binary-feature recursive method in appendix E.2.

## 5. Synergy Distribution in Gradient-Based Methods

A critical aspect of binary feature methods like the Shapley method is that they treat all features in a synergy function as equal contributors to the function output. For example, consider the synergy function of $S = \{1, 2\}$ given by $F(x_1, x_2) = (x_1 - x_1')^{100}(x_2 - x_2')$. $F$ evaluated at $x = (x_1' + 2, x_2' + 2)$ yields $F(x) = 2^{100}2^1 = 2^{101}$. The Shapley method applied to $F$ treats both inputs as equal contributors, and would indicate that $x_1$ and $x_2$ each contributed $\frac{2^{101}}{2}$ to the function increase from the baseline. This

---

[5]Harsanyi (1963) observed Eq. (6) and (7) in the binary feature setting with Möbius transforms. Here we state the continuous input form with synergy functions.

assertion seems unsophisticated, not to mention intuitively incorrect, given we know the mechanism of the interaction function.

The IG exhibits the potential advantages of gradient-based attribution methods by providing a more sophisticated attribution. For $m \in \mathbb{N}^n$, define $(x - x')^m = (x_1 - x_1')^{m_1} \cdots (x_n - x_n')^{m_n}$, taking the convention that if $m_i = 0$ and $x_i = x_i'$, then $(x_i - x_i')^{m_i} = 1$. Define $m! = m_1! \cdots m_n!$, and define $D^m F = \frac{\partial^{\|m\|_1} F}{\partial x_1^{m_1} \cdots \partial x_n^{m_n}}$. We notate the non-constant features of $x^m$ by $S_m = \{i | m_i > 0\}$.

We call a function of the form $F(y) = (y - x')^m$ a monomial centered at $x'$, and note that any monomial centered at an assumed baseline $x'$ is a synergy function of $S_m$. Assuming $m_i > 0$ and taking $x' = 0$, the IG attribution to $y^m$, a synergy function of $S_m$, is:

$$\text{IG}_{\{i\}}(x, y^m) = x_i \int_0^1 m_i(tx)^{(m_1, \ldots, m_i - 1, \ldots, m_n)} dt$$

$$= x_i \int_0^1 m_i t^{\sum m_i - 1} x^{(m_1, \ldots, m_i - 1, \ldots, m_n)} dt$$

$$= m_i x^m \frac{t^{\sum m_j}}{\sum m_j} \Big|_0^1 = \frac{m_i}{\|m\|_1} x^m$$

This means that IG distributes the function change of $F(y) = y^m$ to $x_i$ in proportion to $m_i$. For example, the IG's attribution to our previous problem is $\text{IG}((2, 2), x_1^{100} x_2) = (\frac{100}{101} 2^{101}, \frac{1}{101} 2^{101})$, a solution that seems much more equitable than the Shapley value. Thus the IG can distinguish between features based on the form of the synergy, unlike the Shapley value, which treats all features in a synergy functions as equal contributors.

### 5.1. Continuity Condition

We now move to more rigorously develop the connection between gradient-based methods and monomials. To connect the action of attributions and interactions on monomials to broader functions, we now move towards defining the notion of an interaction being continuous in $F$. Let $\mathcal{C}^\omega$ denote the set of functions that are real-analytic on $[a, b]$. It is well known that any $F \in \mathcal{C}^\omega$ admits to a convergent multivariate Taylor Expansion centered at $x'$:

$$F(x) = \sum_{m \in \mathbb{N}^n} \frac{D^m F(x')}{m!} (x - x')^m \quad (8)$$

Functions in $C^w$ have continuous derivatives of all orders, and those derivatives are bounded in $[a, b]$. Thus, $C^\omega$ it is a well-behaved class that gradient-based interactions ought to be able to assess.

Recall that the Taylor approximation of order $l$ centered at $x'$, denoted $F_l$, is given by:

$$T_l(x) = \sum_{m \in \mathbb{N}^n, \|m\|_1 \leq l} \frac{D^m(F)(x')}{m!} (x - x')^m \quad (9)$$

The Taylor approximation for analytic functions has the property that $D^m T_l$ uniformly converges to $D^m F$ for any $m \in \mathbb{N}^n$ and $x \in [a, b]$. Given this fact, it would be natural

to require that for a given $k^{\text{th}}$-ordered interaction $\mathrm{I}^k$ defined for $C^w$ functions, $\lim_{l \to \infty} \mathrm{I}^k(T_l) = \mathrm{I}^k(F)$. This notion is further justified by the fact that many ML models can be approximated to arbitrary precision by replacing ReLU and max with the parameterized softplus and smoothmax functions, respectively. With this, we propose a continuity axiom requiring interactions for a sequence of Taylor approximations of $F$ to converge to the interactions at $F$.

6. **Continuity of Taylor Approximation for Analytic Functions**: If $\mathrm{I}^k$ is defined for all $(x, x', F) \in [a,b] \times [a,b] \times C^\omega$, then for any $F \in C^\omega$, $\lim_{l \to \infty} \mathrm{I}^k(x, x', T_l) = \mathrm{I}^k(x, x', F)$, where $T_l$ is the $l^{\text{th}}$ order Taylor approximation of $F$ centered at $x'$.

From this we have the following result, who's proof can be found in Appendix E.3:

**Theorem 2.** Let $\mathrm{I}^k$ be an interaction method defined on $[a,b] \times [a,b] \times C^\omega$ which satisfies linearity and continuity of Taylor approximation for analytic functions. Then $\mathrm{I}^k(x, x', F)$ is uniquely determined by the the values $\mathrm{I}^k$ takes for the inputs in the set $\{(x, x', F) : F(y) = (y - x')^m, m \in \mathbb{N}^n\}$.

In section 4 we saw that binary feature methods distribute synergy functions according to a rule, and that rule characterized the method as a whole. Gradient-based methods satisfying linearity and the continuity condition are characterized by their actions on specific sets of elementary synergy functions, monomials. Thus, given our the continuity condition and linearity, we have collapsed the question of continuous interactions to the question of interactions of monomials centered at $x'$. Specifically, if linearity and continuity are deemed desirable, and a means of distributing polynomials can be chosen, then the entire method is determined for analytic functions. This is illustrated by the following corollary (proof located in Appendix E.4):

**Corollary 3.** IG is the unique attribution method on analytic functions that satisfies linearity, the continuity condition, and acts on the inputs $(x, x', (y - x')^m)$ as in Eq. (8).

### 5.2. Integrated Hessians
Next, we present two gradient-based interaction methods. For $m \in \mathbb{N}^n$, the Integrated Hessian of $F(y) = y^m$ at $x$ is:

$$\mathrm{IH}_{\{i,j\}}(y^m) = \frac{2m_i m_j}{\|m\|_1^2} x^m, \quad \mathrm{IH}_{\{i\}}(y^m) = \frac{m_i^2}{\|m\|_1^2} x^m$$

IH distributes a portion of any pure interaction monomial to all nonempty subsets of features in $S_m$, breaking the baseline test for interactions($k \leq n$). For example, although $F(x_1, x_2, x_3) = x_1 x_2$ is a synergy function of $S = \{1, 2\}$, IH distributes some of $F$ to main effects. This can be remedied by directly distributing single and pairwise synergies, then using IH to distribute monomials involving 3 or more variables. The augmented IH of order $k$ acts on monomial functions as follows:

$$\mathrm{IH}_T^{k*}(y^m) = \begin{cases} x^m & \text{if } T = S_m \\ \frac{M_T^k(m)}{\|m\|_1^k} x^m & \text{if } T \subsetneq S_m, |S_m| > k \\ 0 & \text{else} \end{cases} \quad (10)$$

To explain, $\mathrm{IH}^{k*}$ distributes all monomial synergies of size $\leq k$ to their groups, and distributes monomial synergies of size $> k$ to subgroups of $S_m$ in proportion to $M_T^k(m)$. A full treatment of both is given in appendix E.5.

**Corollary 4.** $\mathrm{IH}^{k*}$ is the unique attribution method on analytic functions that satisfies linearity, the continuity condition, and distributes monomials as in Eq. (10).

### 5.3. Sum of Powers: A Top-Distributing Gradient-Based Method
Previously we outlined a $k^{\text{th}}$-order interaction that distributed synergies larger thatn $k$ to all sub-groups. Now we now present the distribution scheme for a gradient-based $k^{\text{th}}$-order interaction we call **Sum of Powers**.[6] We present only its action on monomials here, and detail the method in Appendix E.6. Sum of Powers distributes a monomial as such:

$$\mathrm{SP}_T^k(y^m) = \begin{cases} x^m & \text{if } T = S_m \\ \frac{1}{\binom{|S_m|-1}{k-1}} \frac{\sum_{i \in T} m_i}{\|m\|_1} x^m & \text{if } T \subsetneq S_m, |T| = k \\ 0 & \text{else} \end{cases}$$
$$(11)$$

The highlight is that Sum of Powers satisfies completeness, null feature, linearity, continuity condition, baseline test for interactions, and is a *top-distributing method*. By top-distributing we mean that it projects all synergies larger than the largest available size, $k$, to the largest groups available. This results in Sum of Powers emphasising interactions between features of size $k$, which may be an advantage or disadvantage, depending on the goal of the interaction. We present a corollary below; for full details of the Sum of Powers method, see Appendix E.6.

**Corollary 5.** Sum of Powers is the unique attribution method on analytic functions that satisfies linearity, the continuity condition, and distributes monomials as in Eq. (11).

## 6. Empirical Evaluation
In this section, we compare the performance of the $2^{\text{nd}}$-order Sum of Powers and the unaltered Integrated Hessian methods on a protein tertiary structure dataset. Particularly, we use the Physicochemical Properties of Protein Tertiary Structure dataset from the UCI machine learning repository (Rana, 2013). This dataset consists of 45,730 samples with 9 input features describing the molecular structure of proteins, and the target variable is the size of the residue. For this regression task, we utilize a 2-layer neural network with SoftPlus activation. We run each method on 200 samples. More details about the experiments and additional results are provided in Appendix F.

Figures 1 and 2 report average values for IH and SP, with main effects on the diagonal. We see that both methods

report a strong negative interaction between features 1 and 6, with SP reporting a more negative interaction by 8 points. In the main effects, we see that SP gives more largely positive values for features 1 and 6, while IH is more diminished.

Why is this? Understanding the theory of distributing synergies helps us understand these differences. Theoretically SP reports pure main effects as they are, and all other interactions are projected down to the pairwise interactions. Sum of powers indicates that the pure main effects of features 1 and 6 are positive. IH intermixes main effects and higher order interactions. Since IH's main effects are lower, this means that the pure positive main effects of 6.1 and 9.3 (as seen in SP) are being lowered by generally negative higher-order interactions when IH reports them. A consequence of this is that IH also has a smaller report of the interactions between features 1 and 6: the negative interactions involving features 1 and 6 are being broken up and some are being distributed to main effects, diminishing the report. This strengthening of pairwise interactions is further confirmed by a box-and-whiskers plot (Fig. 3), which shows that SP gives more largely negative values at Q1, 2 and 3.

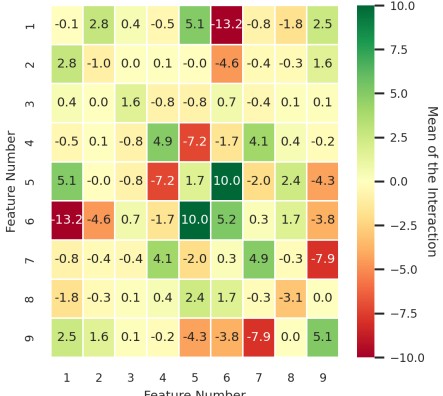

Figure 1. Mean of the Integrated Hessian interaction values.

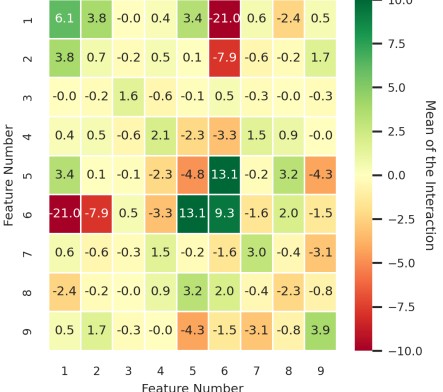

*Figure 2.* Mean of the Sum of Powers interaction values.

Interestingly, Figure 4 also indicates a more pure relationship between features 1 and 6. It is theorized that IH can have wide ranges of coefficients when distributing a monomial (the $M_T^k(m)$ term), while sum of powers is relatively more stable.

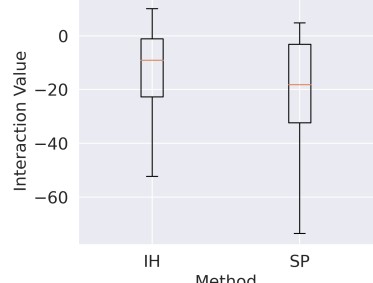

*Figure 3.* Box plot of interaction values of feature 1 and feature 6. Several values with extreme positive and negative interaction values are removed for a cleaner plot.

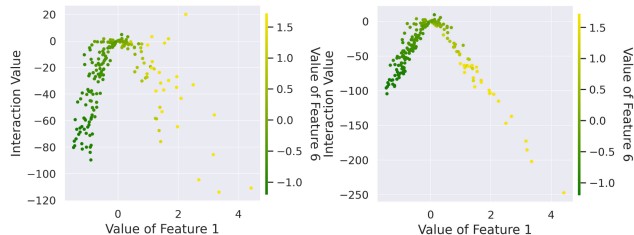

*Figure 4.* Interaction of feature 1 and feature 6. Left: driven by Integrated Hessian. Right: driven by Sum of Powers. X-axis: Feature 1. Y-axis: Interaction value. Colorbar: Feature 6.

## 7. Concluding Remarks

The paradigm of synergy distribution is a useful concept for the analysis and development of attribution and interaction methods, particularly in mission-critical applications such as the ones that appear in health and medicine. First, it can point out weaknesses in existing methods such as the Integrated Hessian and indicate improvements, second, it can lead to new methods such as the Sum of Powers method, and last, it allows new characterization results based on synergy or monomial distribution. As seen in the comparison of Shapley Value vs Integrated Gradient, synergy distribution can play an important role implicitly even when not explicitly discussed in the literature. However, the application of this analysis tool does not settle the question, "which method is best?" There exists conflicting groups of axioms and various combinations of them produce unique interactions. The choice of whether to use a top-distributing or recursively defined method, a binary features or gradient-based method, or some other method may vary with the goal. In the authors' opinion, the possibility of the existence of one "best" method is improbable as various combinations of different axioms lead to the development of unique methods. Thus, choosing methods based on the context of the application seems a more logical approach. Indeed, the existence of unique methods with individual strengths is already studied in game-theoretic cost-sharing literature[7].

---

[7]See the Shapley value vs Aumann-Shapley value vs serial cost for cost-sharing (Friedman & Moulin, 1999), or the Shapley vs Banzhaf interaction indices (Grabisch & Roubens, 1999).

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

# Appendix

## A. Table of Methods

All listed methods satisfy completeness, linearity, null feature, and symmetry. All gradient-based methods satisfy the continuity condition. All interaction methods also satisfy baseline test for interactions ($k \leq n$) unless otherwise noted. We do not list interaction distribution, which is a combination of baseline test for interactions ($k \leq n$) and being top-distributing in the binary features scheme.

## B. Table of Methods

All listed methods satisfy completeness, linearity, null feature, and symmetry. All gradient-based methods satisfy the continuity condition. All interaction methods also satisfy baseline test for interactions ($k \leq n$) unless otherwise noted. We do not list interaction distribution, which is a combination of baseline test for interactions ($k \leq n$) and being top-distributing in the binary features scheme.

| Name | Properties | Distribution Rule |
|---|---|---|
| Synergy Function | unique $n^{\text{th}}$-order interaction | $\phi_T(\Phi_S)(x) = \begin{cases} \Phi_S(x) & \text{if } S = T \\ 0 & \text{if } S \neq T \end{cases}$ |
| Shapley Value | attribution method binary features | $\text{Shap}_i(x, \Phi_S) = \begin{cases} \frac{\Phi_S(x)}{|S|} & \text{if } i \in S \\ 0 & \text{if } i \notin S \end{cases}$ |
| Integrated Gradients | attribution method gradient-based | $\text{IG}_i(x, y^m) = \begin{cases} \frac{m_i}{\|m\|_1} x^m & \text{if } i \in S_m \\ 0 & \text{if } i \notin S_m \end{cases}$ |
| Shapley-Taylor | binary features top-distributing | $\text{ST}_T^k(x, \Phi_S) = \begin{cases} \Phi_S(x) & \text{if } T = S \\ \frac{\Phi_S(x)}{\binom{|S|}{k}} & \text{if } T \subsetneq S, |T| = k \\ 0 & \text{else} \end{cases}$ |
| Sum of Powers | gradient-based top-distributing | $\text{SP}_T^k(x, y^m) = \begin{cases} x^m & \text{if } T = S_m \\ \frac{\sum_{i \in T} m_i}{\binom{|S_m|-1}{k-1}\|m\|_1} x^m & \text{if } T \subsetneq S_m, |T| = k \\ 0 & \text{else} \end{cases}$ |
| Recursive Shapley | binary features iterative breaks baseline test | $\text{RS}_T^k(x, \Phi_S) = \begin{cases} \frac{N_T^k}{|S|^k} \Phi_S(x) & \text{if } T \subseteq S \\ 0 & \text{else} \end{cases}$ |
| Augmented Recursive Shapley | binary features iterative | $\text{RS}_T^{k*}(x, \Phi_S) = \begin{cases} \Phi_S(x) & \text{if } T = S \\ \frac{N_T^k}{|S|^k} \Phi_S(x) & \text{if } T \subsetneq S, |S| > k \\ 0 & \text{else} \end{cases}$ |
| Integrated Hessian | gradient-based iterative breaks baseline test | $\text{IH}_T^k(x, y^m) = \begin{cases} \frac{M_T^k(m)}{\|m\|_1^k} x^m & \text{if } T \subseteq S_m \\ 0 & \text{else} \end{cases}$ |
| Augmented Integrated Hessian | gradient-based iterative | $\text{IH}_T^{k*}(x, y^m) = \begin{cases} x^m & \text{if } T = S_m \\ \frac{M_T^k(m)}{\|m\|_1^k} x^m & \text{if } T \subsetneq S_m, |S_m| > k \\ 0 & \text{else} \end{cases}$ |

## C. Axioms and the Distribution of Synergies

Here we comment on the interplay between axioms and synergy functions. First, we present a version of the baseline test for interactions which applies for $k \leq n$. The idea is a generalization of the ($k = n$) case; that if $\text{I}^k$ is a $k^{\text{th}}$-order interaction and $\Phi_S$ is some pure interaction function with $|S| \leq k$, then $\text{I}^k(\Phi_S)$ should not report interactions for any set but $S$. We give this as an axiom:

7. **Baseline Test for Interactions** ($k \leq n$): For baseline $x'$ and any synergy function $\Phi_S$ with $|S| \leq k$, if $T \subsetneq S$, then $\mathrm{I}_T^k(\Phi_S) = 0$.

This is a weaker version of the defining axiom of Shapley-Taylor (Sundararajan et al., 2020), which states:

8. **Interaction Distribution**: For baseline $x'$ and any synergy function $\Phi_S$, if $T \subsetneq S$ and $|T| < k$, then $\mathrm{I}_T^k(\Phi_S) = 0$.

The baseline test of interactions asserts that if a synergy function is for a group of at least size $k$, $\mathrm{I}^k$ should not report interactions for any other group. The interaction distribution asserts the same, and adds the caveat that if the synergy function is for a group of size larger than $k$, it must be distributed only to groups of size $k$.

We now detail how some of these axioms can be formulated as constraints on the distribution of synergies.

1. Completeness: enforces that any method distributes a synergy among sets of inputs. Formally, for a synergy function $\Phi_S$, we may say that $\mathrm{I}_T^k(x, \Phi_S) = w_T(x, \Phi_S) \times \Phi_S(x)$, where $w_T$ is some function satisfying $\sum_{T \subseteq \mathcal{P}_k} w_T(x, \Phi_S) = 1$.

2. Linearity: enforces that $\mathrm{I}^k(F)$ is the sum of $\mathrm{I}^k$ applied to the synergies of $F$. Formally, $\mathrm{I}^k(F) = \sum_{T \subset \mathcal{P}_k} \mathrm{I}^k(\phi_T(F))$.

3. Null Feature: enforces that $\mathrm{I}^k$ only distributed $\Phi_S$ to groups $T \subseteq S$.

4. Baseline Test for Interaction($k \leq n$): enforces that $\Phi_S$ is not distributed to groups $T \subsetneq S$ when $|S| \leq k$.

5. Interaction Distribution: enforces that $\Phi_S$ is not distributed to groups $T \subsetneq S$ when $|S| \leq k$, and is distributed only to groups of size $k$ when $|S| > k$.

6. Symmetry[8]: enforces that a synergy $\Phi_S$ be distributed equally among groups in the binary features case.

### C.1. Statement of Symmetry Axiom

Let $\pi$ be an ordering of the features in $N$. We loosely quote the definition of symmetry from Sundararajan et al. (2020), altering the binary feature setting to a continuous feature setting:

7. *Symmetry Axiom: for all $F \in \mathcal{F}$, for all permutations $\pi$ on $N$:*

$$I_S^k(x, x', F) = I_{\pi S}^k(\pi x, \pi x', F \circ \pi^{-1}), \tag{12}$$

*where $\circ$ denotes function composition, $\pi S := \{\pi(i) : i \in S\}$, and $(\pi x)_{\pi(i)} = x_i$.*

This axioms implies that if we relabel the features, then interactions for the relabeled features will concur with interactions before relabeling. It requires that the domain, $[a, b]$, is closed under permutations of inputs, meaning it is of the form $[a_1, b_1]^n$.

## D. Synergy Function

### D.1. Proof of Theorem 1

*Proof.* Let I be any $n$-ordered interaction that satisfies the given axioms, and let $x, x' \in [a, b] \times [a, b]$ be arbitrarily chosen. We assume that all interactions are taken with respect to input $x$ and baseline $x'$. For ease of notation, we define $F_S(x) = F(x_S)$ for $F \in \mathcal{F}(x, x')$.

For any nonempty $S \in \mathcal{P}_n$, note that $\mathrm{I}_S(F) = \mathrm{I}_S(F - F_S + F_S)$. Note that $(F - F_S)(x_S)$ is constant. Thus, $\mathrm{I}_S(F - F_S) = 0$ for any $S \in \mathcal{P}_k$ by the baseline test for interaction. Thus, by linearity of zero-valued functions, we have established that $\mathrm{I}_S(F) = \mathrm{I}_S(F_S)$ for any $S \in \mathcal{P}_k$.

We now proceed by strong induction:

---

[8]See appendix C.1 for a statement of symmetry axiom.

$|S| = 1$ case: Let $i \in N$ and choose $F \in \mathcal{F}$. Note that $F_{\{i\}}$ does not vary with any feature but $x_i$. This implies that for $S \neq \{i\}$, $I_S(F_{\{i\}}) = 0$ by null feature. By completeness, $I_{\{i\}}(F_{\{i\}}) = F_{\{i\}}(x) - F_{\{i\}}(x')$, and $I_{\{i\}}(F)$ is uniquely determined. Thus $I_S(F)$ is uniquely determined for $|S| = 1$.

$|S| \leq k \Rightarrow |S| = k + 1$ case: Suppose that for any $G \in \mathcal{F}[a, b]$ and any $S \subseteq \{1, ..., n\}$ such that $|S| \leq k$, $I_S(G)$ is uniquely determined. Let $T \in \mathcal{P}_n, |T| = k + 1$, $F \in \mathcal{F}$. It has been established that $I_T(F) = I_T(F_T)$. Note that for all $S \subsetneq T$, we have $|S| \leq k$, so $I_S(F_T)$ is uniquely determined by the induction hypotheses. Since $F_T$ does not vary in each $x_i$ such that $i \notin T$, we have $I_S(F_T) = 0$ for $S \nsubseteq T$ by null feature. By completeness, $F_T(x) - F_T(x') = \sum_{S \subseteq \mathcal{P}_k} I_S(F_T) = \sum_{S \subseteq T} I_S(F_T)$. Thus $I_T(F_T) = F_T(x) - F_T(x') - \sum_{S \subsetneq T} I_S(F_T)$. Since $I_T(F) = I_T(F_T)$ equals the sum of uniquely determined terms, $I_T(F)$ is uniquely determined. $\square$

## D.2. Context of Synergy Function

The properties of the synergy function stem from properties of the Möbius transform. Specifically, because the synergy function is defined by the Möbious Transform, it inherits many of its properties, including completeness, null feature, linearity of zero-valued functions, and baseline test for interactions ($n = k$). The primary precursor to the synergy function is the Harsanyi dividend (Harsanyi, 1963), which is like the Möbius transform and is formulated for discrete-input settings. More recently, the Shapley-Taylor Interaction Index (Sundararajan et al., 2017) and Faith-Shap (Tsai et al., 2022) take the form of the Möbius Transform when $k = n$. The novelty of the synergy function is that, while previous works assumed $F$ to be a set function (as in section 2.4), the synergy function is a linear functional between continuous input functions. Consequently, Corollary 1 is novel, not only because of the inclusion of baseline test for interactions ($k = n$), but also because all axioms do not assume $F$ is a set function.

## D.3. Proof of Corollary 1

We proceed to show the synergy function satisfies completeness, linearity, null feature, and baseline test for interactions ($k \leq n$).

*Proof.* **Completeness**: For any $v : \{0, 1\}^n \to \mathbb{R}$, Sundararajan et al. (2020, Appendix 7.1) shows that the Möbius transform has the property that,

$$v(T) = \sum_{S \subseteq T} a(v)(S). \tag{13}$$

Using this, observe,

$$F(x') + \sum_{S \in \mathcal{P}_n} \phi_S(F)(x) = \sum_{S \subseteq N} a(F(x_{(\cdot)}))(S)$$
$$= F(x_N) \tag{14}$$
$$= F(x),$$

which established completeness.

**Linearity of Zero-Valued Functions**: We simply establish $\phi$ is linear.

$$\phi_S(cF + dG)(x) = a(cF(x_{(\cdot)}) + dG(x_{(\cdot)}))(S)$$
$$= \sum_{T \subseteq S} (-1)^{|S|-|T|} \left[ (cF(x_{(\cdot)}) + dG(x_{(\cdot)}))(T) \right]$$
$$= c \sum_{T \subseteq S} (-1)^{|S|-|T|} F(x_{(\cdot)})(T) + d \sum_{T \subseteq S} (-1)^{|S|-|T|} G(x_{(\cdot)})(T) \tag{15}$$
$$= c\phi_S(F)(x) + d\phi_S(G)(x)$$

**Baseline Test for Interactions**: Suppose $F(x_S)$ is constant.

$$
\begin{aligned}
\phi_S(F)(x) &= a(F(x_{(\cdot)}))(S) \\
&= \sum_{T \subseteq S} (-1)^{|S|-|T|} F(x_T) \\
&= \sum_{T \subseteq S} (-1)^{|S|-|T|} F(x') \\
&= F(x') \sum_{0 \le i \le |S|} \binom{|S|}{i} (-1)^{|S|-i} \\
&= 0
\end{aligned}
\tag{16}
$$

**Null Feature**: Suppose $F$ does not vary in some $x_i$ and $i \in S$. Then,

$$
\begin{aligned}
\phi_S(F)(x) &= a(F(x_{(\cdot)}))(S) \\
&= \sum_{T \subseteq S} (-1)^{|S|-|T|} F(x_T) \\
&= \sum_{T \subseteq S, i \in T} (-1)^{|S|-|T|} F(x_T) + \sum_{T \subseteq S, i \notin T} (-1)^{|S|-|T|} F(x_T) \\
&= \sum_{T \subseteq S \setminus \{i\}} (-1)^{|S|-(|T|+1)} F(x_{T \cup \{i\}}) + \sum_{T \subseteq S \setminus \{i\}} (-1)^{|S|-|T|} F(x_T) \\
&= - \sum_{T \subseteq S \setminus \{i\}} (-1)^{|S|-|T|)} F(x_T) + \sum_{T \subseteq S \setminus \{i\}} (-1)^{|S|-|T|} F(x_T) \\
&= 0
\end{aligned}
\tag{17}
$$

$\square$

### D.4. Proof of Corollary 2

*Proof.* We proceed in the order given in Corollary 2.

**1. Pure interaction sets are disjoint, meaning $C_S \cap C_T = \emptyset$ whenever $S \ne T$.**

Suppose $S, T \in \mathcal{P}_n$ with $T \ne S$. We proceed by contradiction and suppose $F \in C_S \cup C_T$. WLOG $\exists i \in S \setminus T$, implying that $F$ varies in feature $i$ since $F$ is a synergy function of $S$, and $F$ does not vary in feature $i$, since $F$ is a synergy function of $T$. This is a contradiction. Thus $C_S \cap C_T = \emptyset$.

**2. $\phi_S$ projects $\mathcal{F}$ onto $C_S \cup \{0\}$. That is, $\phi_S(F) \in C_S \cup \{0\}$ and $\phi_S(\phi_S(F)) = \phi_S(F)$**

Let $F \in \mathcal{F}$. First, for the degenerate case, $\phi_\emptyset(F) = F(x')$, which is a constant function. For any constant $c$, $\phi_\emptyset(c) = c$, implying $\phi_\emptyset$ is a projection and surjective for the range $C_\emptyset \cup \{0\}$. Thus $\phi_\emptyset$ projects $\mathcal{F}$ onto $C_\emptyset \cup \{0\}$.

Now we will show that $\phi_S(F)$ either is a pure interaction of $S$ or is 0 in the non-degenerate case. Suppose $x_i = x_i'$ for some $i \in S$. Then,

$$\phi_S(F)(x) = \sum_{T \subseteq S} (-1)^{|S|-|T|} F(x_T)$$

$$= \sum_{T \subseteq S, i \in T} (-1)^{|S|-|T|} F(x_T) + \sum_{T \subseteq S, i \notin T} (-1)^{|S|-|T|} F(x_T)$$

$$= \sum_{T \subseteq S \setminus \{i\}} (-1)^{|S|-(|T|+1)} F(x_{T \cup \{i\}}) + \sum_{T \subseteq S \setminus \{i\}} (-1)^{|S|-|T|} F(x_T)$$

$$= - \sum_{T \subseteq S \setminus \{i\}} (-1)^{|S|-|T|)} F(x_T) + \sum_{T \subseteq S \setminus \{i\}} (-1)^{|S|-|T|} F(x_T)$$

$$= 0$$

Thus $\phi_S(F) = 0$ whenever $x_i = x_i'$ for some $i \in S$, and $\phi_S(F)$ satisfies condition 1 for being a pure interaction of $S$.

Now, inspecting the definition, $\phi_S(F)(x) = \sum_{T \subseteq S}(-1)^{|S|-|T|} F(x_T)$, so $\phi_S(F)$ does not vary in $x_i$, $i \notin S$. Lastly, suppose that $F$ does not vary in some $x_i$, $i \in S$. Since $\phi$ satisfies null feature, $\phi_S(F) = 0$. So either $\phi_S(F)$ varies in all $x_i$ such that $i \in S$, or $\phi_S(F) = 0$. If the former, $\phi_S(F)$ satisfies condition 2 for being a pure interaction of $S$; if the latter, $\phi_S(F) = 0$. Thus $\phi_S(F) = 0$ or $\phi_S(F)$ is a pure interaction function of $S$, implying the range of $\phi_S$ is $C_S \cup \{0\}$.

Now let $\Phi_S \in C_S$. Note

$$\phi_S(\Phi_S)(x) = \sum_{T \subseteq S} (-1)^{|S|-|T|} \Phi_S(x_T)$$

$$= \sum_{T=S} (-1)^{|S|-|T|} \Phi_S(x_T)$$

$$= \Phi_S(x_S)$$

$$= \Phi_S(x)$$

It is plain by the definition that $\phi_S(0) = 0$. Thus $\phi_S$ is surjective for the range $C_S \cup \{0\}$. Since the range of $\phi_S$ is $C_S \cup \{0\}$, $\phi$ maps elements of $C_S$ to themselves, and maps 0 to 0, so $\phi_S$ is a projection.

**3. For $\Phi_T \in C_T$, we have $\phi_S(\Phi_T) = 0$ whenever $S \neq T$.**

Let $\Phi_T \in C_T$ and $T \neq S$. If $\exists i \in S \setminus T$, then $\phi_S(\Phi_T) = 0$ by null feature. Otherwise $S \subsetneq T$, and $\phi_S(\Phi_T) = 0$ be baseline test for interactions ($k = n$).

**4. $\phi$ uniquely decomposes $F \in \mathcal{F}$ into a set of pure interaction functions on distinct groups of features. That is, there exists $\mathcal{P} \subset \mathcal{P}_n$ such that $F = \sum_{S \in \mathcal{P}} \Phi_S$, where each $\Phi_S \in C_S$. Further more, only one such representation exists, $\Phi_S = \phi_S(F)$ for each $S \in \mathcal{P}$, and $\phi_S(F) = 0$ for each $S \in \mathcal{P}_n \setminus \mathcal{P}$.**

$F = \sum_{S \in \mathcal{P}_n} \phi_S(F)$, and each $\phi_S(F) \in C_S \cup \{0\}$. Since $0 + \phi_\emptyset(F) \in C_\emptyset$ and we may gather all the $\phi_S(F)$ terms that are zero into the $C_\emptyset$ term, we have shown a decomposition exists.

Let it be that $F(x) = \sum_{S \in \mathcal{P}} \Phi_S(x)$ for some $\mathcal{P} \in \mathcal{P}_n$, where each $\Phi_S$ is an interaction function in $S$. By the results already established, we have for any $T \in \mathcal{P}$

$$\phi_S(F) = \phi_S\left(\sum_{T \in \mathcal{P}} \Phi_T\right)$$

$$= \sum_{T \in \mathcal{P}} \phi_S(\Phi_T)$$

$$= \phi_S(\Phi_S)$$

$$= \Phi_S$$

If $S \notin \mathcal{P}$, then

$$\phi_S(F) = \phi_S(\sum_{T \in \mathcal{P}} \Phi_T)$$

$$= \sum_{T \in \mathcal{P}} \phi_S(\Phi_T)$$

$$= 0$$

Now suppose that there are two decompositions, $\sum_{S \in \mathcal{P}^1} \Phi_S^1 = F = \sum_{S \in \mathcal{P}^2} \Phi_S^2$. WLOG suppose $S \in \mathcal{P}^1 \setminus \mathcal{P}^2$. Then $\phi_S(F) = 0$ since $S \notin \mathcal{P}^2$ and $\phi_S(F) = \Phi_S^1$ since $S \in \mathcal{P}^1$. Thus $\Phi_S^1 = 0$ and $S = \emptyset$. Thus $\mathcal{P}^1 \triangle \mathcal{P}^2$ equals either $\emptyset$ or $\{\emptyset\}$, and in the case that $\mathcal{P}^1 \triangle \mathcal{P}^2 = \{\emptyset\}$ the extra term corresponding to $\emptyset$ in one of the sums is 0, and does not effect the decomposition. Now, if $\mathcal{P}^1 \triangle \mathcal{P}^2 = \emptyset$, then for any $S \in \mathcal{P}^1, \mathcal{P}^2$, we have $\Phi_S^1 = \phi_S(F) = \Phi_S^2$. Thus, the decomposition is unique. $\square$

# E. $k^{\text{th}}$-Order Interaction Methods

Here we give an in depth treatment of the Shapley Taylor, Recursive Shapley, Integrated Hessian, and Sum of Powers methods, as well as the augmentations to the recursive methods. We define the methods and show that each method is the unique method that satisfies linearity, their distribution policy, and in the case of gradient methods, the continuity condition. We also prove that each method satisfies desirable properties such as completeness, null feature, symmetry, and, if applicable, baseline test for interactions ($k \leq n$).

## E.1. The Shapley-Taylor Interaction Index

Several $k^{\text{th}}$-order interactions that extend Shapley values have been proposed, all of which are binary feature methods (Grabisch & Roubens, 1999),(Tsai et al., 2022). Here we focus our analysis on the Shapley-Taylor method (Sundararajan et al., 2020). First, define $\delta_{S|T} F(x) = \sum_{W \subseteq S} (-1)^{|S|-|W|} F(x_{W \cup T})$, which measures the marginal impact of including the features in $S$ when the features in $T$ are already present based on the inclusion-exclusion principle. The **Shapley-Taylor Interaction Index** of order $k$ (Sundararajan et al., 2020) is then given by:

$$\text{ST}_S^k(x, F) = \begin{cases} \frac{k}{n} \sum_{T \subseteq N \setminus S} \frac{\delta_{S|T} F(x)}{\binom{n-1}{|T|}} & \text{if } |S| = k \\ \delta_{S|\emptyset}(F) & \text{if } |S| < k. \end{cases} \quad (18)$$

Shapley-Taylor prioritizes interactions of order $k$ and its unique contribution is to satisfy the interaction distribution axiom, which is discussed in Appendix C.

### E.1.1. Analysis of Shapley-Taylor using Synergies

For a synergy function $\Phi_S$, the Shapley-Taylor interaction index of order $k$ for a group of features $T \in \mathcal{P}_k$ is given by:

$$\text{ST}_T^k(\Phi_S) = \begin{cases} \Phi_S(x) & \text{if } T = S \\ \frac{\Phi_S(x)}{\binom{|S|}{k}} & \text{if } T \subsetneq S, |T| = k \\ 0 & \text{else} \end{cases} \quad (19)$$

The Shapley-Taylor distributes each synergy function of $S$ to its group, unless is too large ($|S| > k$), in which case it distributes the synergy equally among all subsets of $S$ of size $k$. This type of method is top-distributing, as every synergy function of a group $T$, $|T| > k$, is distributed only to groups of order $k$.

As with the Shapley value, the Shapley-Taylor is characterized by this action on synergy functions:

**Proposition 2.** *(Sundararajan et al., 2020, Prop 4)* The Shapley–Taylor Interaction Index of order $k$ is the unique $k^{\text{th}}$-order interaction index that satisfies linearity and acts on synergy functions as in Eq. (19).

**E.2. Recursive Shapley and Augmented Recursive Shapley**

There is another binary feature $k^{\text{th}}$-order interaction method similar to Shapley-Taylor, briefly motioned in Sundararajan et al. (2020), with the distinction that it is not top-distributing. Here we detail and augment the method. Similarly to the Integrated Hessian, we may take the Shapley value recursively to gain pairwise interaction between $x_i$ and $x_j$, given by $\text{RS}_{\{i,j\}}(x, F) = \text{Shap}_i(x, \text{Shap}_j(\cdot, F)) + \text{Shap}_j(x, \text{Shap}_i(\cdot, F)) = 2\text{Shap}_i(x, \text{Shap}_j(\cdot, F))$. Main effects for $x_i$ would be $\text{Shap}_i(x, \text{Shap}_i(\cdot, F))$.

More generally, consider expanding the expression $\|y\|_1^k$, and let $N_T^k$ denote the sum of coefficients associated exactly with the variables with indices in $T$. Then the **Recursive Shapley** of order $k$ distributes synergy functions as such:

$$\text{RS}_T^k(\Phi_S) = \begin{cases} \frac{N_T^k}{|S|^k} \Phi_S(x) & \text{if } T \subseteq S \\ 0 & \text{else} \end{cases}, \tag{20}$$

where in the case $T = S = \emptyset$ we set $\frac{N_T^k}{|S|^k} := 1$. This formulation, however, has the disadvantage of distributing a portion of synergy functions for groups sized $\leq k$ to subgroups. For example, the recursively Shapley reports that a synergy function $\Phi_{\{1,2,3\}}(x)$ also has interactions for subgroup $\{1,2\}$. This violates the baseline test for interactions ($k \leq n$). We can modify the method to avoid this issue, causing Recursive Shapley to satisfy the baseline test for interactions ($k \leq n$) axiom. We explicitly detail the Recursive Shapley and modification in E.2. We also give the following Theorem (Proof in Appendix E.2.2):

**Theorem 3.** The Recursive Shapley of order $k$ is the unique $k^{\text{th}}$-order interaction index that satisfies linearity and acts on synergy functions as in Eq. (20).

E.2.1. DEFINING RECURSIVE SHAPLEY

Here we detail the properties of Recursive Shapley and Augmented Recursive Shapley. Let $\sigma_T^k$ be the set of sequences of length $k$ such that the sequence is made of the elements of $T \neq \emptyset$ and each element appears at least once. For example, $\sigma_{\{1,2\}}^3 = \{(1,1,2),(1,2,1),(1,2,2),(2,1,1),(2,1,2),(2,2,1)\}$. Calculating the size of $\sigma_T^k$, $|\sigma_T^k| = \sum_{|l|=k \text{ s.t. } S_l=T} \binom{k}{l} = N_T^k$. For a given sequence $s$, define $\text{IG}_t(x, F)$ be a recursive implementation of the Shapley method according to the sequence $s$, i.e., $\text{Shap}_{(1,2,3)}(x, F) = \text{Shap}_3(x, \text{Shap}_2(\cdot, \text{Shap}_1(\cdot, F)))$. We can then define the $k^{\text{th}}$-order Recursive Shapley for $T \neq \emptyset$ as:

$$\text{RS}_T^k(x, F) = \sum_{s \in \sigma_T^k} \text{Shap}_s(x, F) \tag{21}$$

and define $\text{RS}_\emptyset^k(x, x', F) := F(x')$.

We now move to inspect this equation and establish some properties. Eq. (6) states that for a synergy function $\Phi_S$, $S \neq \emptyset$,

$$\text{Shap}_i(x, \Phi_S) = \begin{cases} \frac{\Phi_S(x)}{|S|} & \text{if } i \in S \\ 0 & \text{if } i \notin S \end{cases} \tag{22}$$

Then for a given sequence $s \in \sigma_T^k$ and synergy function $\Phi_S$, if $T \subseteq S$ then,

$$
\begin{aligned}
\text{Shap}_s(x, \Phi_S) &= \text{Shap}_{s_k}(x, \text{Shap}_{s_{k-1}}(...\text{Shap}_{s_1}(\cdot, \Phi_S)....) \\
&= \text{Shap}_{s_k}(x, \text{Shap}_{s_{k-1}}(...\text{Shap}_{s_2}(\cdot, \frac{\Phi_S}{|S|})....) \\
&= \text{Shap}_{s_k}(x, \text{Shap}_{s_{k-1}}(...\text{Shap}_{s_3}(\cdot, \frac{\Phi_S}{|S|^2})....) \\
&= ... \\
&= \text{Shap}_{s_k}(x, \frac{\Phi_S}{|S|^{k-1}})) \\
&= \frac{\Phi_S(x)}{|S|^k}
\end{aligned}
\tag{23}
$$

However, if $T \subsetneq S$ then there exists an element of $s$ that is not in $S$, and:

$$\text{Shap}_s(x, \Phi_S) = 0, \tag{24}$$

due to some $s_j \notin S$ in the sequence.

### E.2.2. RECURSIVE SHAPLEY'S DISTRIBUTION POLICY

Now, to show how Recursive Shapley distributes synergies, apply the definition of recursive Shapely for $S \neq \emptyset$ to get:

$$
\begin{aligned}
\text{RS}_T^k(x, \Phi_S) &= \sum_{s \in \sigma_T^k} \text{Shap}_s(x, \Phi_S) \\
&= \begin{cases} \sum_{s \in \sigma_T^k} \frac{\Phi_S(x)}{|S|^k} & \text{if } T \subseteq S \\ \sum_{s \in \sigma_T^k} 0 & \text{if } T \nsubseteq S \end{cases} \\
&= \begin{cases} \frac{N_T^k}{|S|^k} \Phi_S(x) & \text{if } T \subseteq S \\ 0 & \text{if } T \nsubseteq S \end{cases}
\end{aligned}
\tag{25}
$$

We also gain the above for $S = \emptyset$ by setting $\frac{N_T^k}{|S|^k} = 1$ when $T = \emptyset$. This establishes the distribution scheme in Eq. (20).

Recursive Shapley is also underline{linear} because it it the sum of function compositions of composition of linear functions. This establishes Theorem 3.

### E.2.3. PROPERTIES OF RECURSIVE SHAPLEY

To show Recursive Shapley satisfies completeness, observe for $S \neq \emptyset$:

$$
\begin{aligned}
\sum_{T \in \mathcal{P}_k, |T| > 0} \text{RS}_T^k(x, \Phi_S) &= \sum_{T \subseteq S} N_T^k \frac{\Phi_S(x)}{|S|^k} \\
&= \frac{\Phi_S(x)}{|S|^k} \sum_{T \subseteq S} N_T^k \\
&= \frac{\Phi_S(x)}{|S|^k} |S|^k \\
&= \Phi_S(x)
\end{aligned}
\tag{26}
$$

The case when $S = \emptyset$ is easily verified by inspecting the synergy distribution policy of RS.

To show Recursive Shapley satisfies null feature, suppose that $F$ does not vary in $x_i$. Then for any $S \in \mathcal{P}_k$ such that $i \in S$, $\phi_S(F) = 0$ since the synergy function is an interaction satisfying null feature. Then if $i \in T$,

$$
\begin{aligned}
\text{RS}_T^k(x, F) &= \sum_{S \in \mathcal{P}_k} \text{RS}_T^k(x, \phi_S(F)) \\
&= \sum_{S \in \mathcal{P}_k \text{ s.t. } i \in S} \text{RS}_T^k(x, \phi_S(F)) + \sum_{S \in \mathcal{P}_k \text{ s.t. } i \notin S} \text{RS}_T^k(x, \phi_S(F)) \\
&= \sum_{S \in \mathcal{P}_k \text{ s.t. } i \in S} \text{RS}_T^k(x, 0) + \sum_{S \in \mathcal{P}_k \text{ s.t. } i \notin S} 0 \\
&= 0
\end{aligned}
\tag{27}
$$

Where the terms in the second sum are zero by Eq. (20).

To show Recursive Shapley satisfies symmetry, let $\pi$ be a permutation on $N$. Note that for $\Phi_S \in C_S$, we have $\Phi_S \circ \pi^{-1}$ is a pure interaction function in $\pi S$ with baseline $\pi x'$. Then

$$
\begin{aligned}
\mathrm{RS}^k_{\pi T}(\pi x, \pi x', \Phi_S \circ \pi^{-1}) &= \begin{cases} \frac{N^k_{\pi T}}{|\pi S|^k} \Phi_S \circ \pi^{-1}(\pi x) & \text{if } \pi T \subseteq \pi S \\ 0 & \text{if } \pi T \nsubseteq \pi S \end{cases} \\
&= \begin{cases} \frac{N^k_T}{|S|^k} \Phi_S(x) & \text{if } T \subseteq S \\ 0 & \text{if } T \nsubseteq S \end{cases} \\
&= \mathrm{RS}^k_T(x, x', \Phi_S)
\end{aligned}
$$

So RS is symmetric on synergy functions. Now use the synergy decomposition of $F \in \mathcal{F}$ to show RS is generally symmetric.

E.2.4. AUGMENTED RECURSIVE SHAPLEY AND PROPERTIES

The synergy function $\phi$ is taken implicitly with respect to a baseline appropriate to $F$. To make the baseline choice explicit, we write $\phi(F) = \phi(x', F)$. Augmented Recursive Shapley is then defined as:

$$
\mathrm{RS}^{k*}_T(x, x', F) = \phi_T(x', F)(x) + \mathrm{RS}^k_T\left(x, x', F - \sum_{S \in \mathcal{P}_k} \phi_S(x', F)\right) \tag{28}
$$

With the above augmentation, $\mathrm{IH}^{k*}$ explicitly distributes synergies $\phi_T(F)$ to group $T$ whenever $|T| \le k$, and distributes higher synergies as $\mathrm{IH}^k$.

The above is a linear function of $F$. Plugging in $\Phi_S$ to the above gains the following distribution policy:

$$
\mathrm{RS}^{k*}_T(\Phi_S) = \begin{cases} \Phi_S(x) & \text{if } T = S \\ \frac{N^k_T}{|S|^k} \Phi_S(x) & \text{if } T \subsetneq S, |S| > k \\ 0 & \text{else} \end{cases} \tag{29}
$$

Because each $F$ ha a unique synergy decomposition, we have

**Corollary 6.** Augmented Recursive Shapley of order $k$ is the unique $k^{\text{th}}$-order interaction index that satisfies linearity and acts on synergy functions as in Eq. (29).

To show that Augmented Recursive Shapley satisfies null feature, let $F$ not vary in some feature $x_i$ and let $i \in T$. Then

$$
\begin{aligned}
\mathrm{RS}^{k*}_T(x, F) &= \sum_{S \in \mathcal{P}_n} \mathrm{RS}^{k*}_T(x, \phi_S(F)) \\
&= \mathrm{RS}^{k*}_T(x, \phi_T(F)) + \sum_{T \subsetneq S, |S| > k} \mathrm{RS}^{k*}_T(x, \phi_S(F)) \\
&= \mathrm{RS}^{k*}_T(x, 0) + \sum_{T \subsetneq S, |S| > k} \frac{N^k_T}{|S|^k} \phi_S(F)(x) \\
&= 0 + \sum_{T \subsetneq S, |S| > k} 0 \\
&= 0
\end{aligned}
$$

Thus Augmented Recursive Shapley satisfies null feature.

To show Augmented Recursive Shapley satisfies baseline test for interactions ($k \le n$), let $T \subsetneq S$, $|S| \le k$, and $\Phi_S \in C_S$. Then $\mathrm{RS}^{k*}_T(x, \Phi_S) = 0$ by Eq.(29).

To show Augmented Recursive Shapley satisfies completeness, consider the synergy function $\Phi_S$. If $|S| \le k$, Eq. (29) shows completeness. If $|S| > k$, then follow the proof of completeness for Recursive Shapley.

To show Augmented Recursive Shapley satisfies symmetry, consider a synergy function $\Phi_S \in C_S$ and permutation $\pi$. Note that for $\Phi_S \in C_S$, we have $\Phi_S \circ \pi^{-1}$ is a pure interaction function in $\pi S$ with baseline $\pi x'$. Then

$$
\mathrm{RS}^{k*}_{\pi T}(\pi x, \pi x', \Phi_S \circ \pi^{-1}) = \begin{cases} \frac{N^k_{\pi T}}{|\pi S|^k} \Phi_S \circ \pi^{-1}(\pi x) & \text{if } \pi T = \pi S \\ \frac{N^k_{\pi T}}{|\pi S|^k} \Phi_S \circ \pi^{-1}(x) & \text{if } \pi T \subsetneq \pi S, |\pi S| > k \\ 0 & \text{else} \end{cases}
$$

$$
= \begin{cases} \frac{N^k_T}{|S|^k} \Phi_S(x) & \text{if } T \subseteq S \\ \frac{N^k_T}{|S|^k} \Phi_S(x) & \text{if } T \subsetneq S, |S| > k \\ 0 & \text{else} \end{cases}
$$

$$
= \mathrm{RS}^{k*}_T(x, x', \Phi_S)
$$

## E.3. Proof of Theorem 2

*Proof.* Let $\mathrm{I}^k$ be a $k^{\text{th}}$-order interaction method defined for all $(x, x', F) \in [a, b] \times [a, b] \times C^\omega$. Fix $x'$ and $x$. Let $T_l$ be the $l^{\text{th}}$ order Taylor approximation of $F$ at $x'$. Then

$$
\mathrm{I}^k(x, x', F) = \lim_{l \to \infty} \mathrm{I}^k(x, x', T_l)
$$

$$
= \sum_{m \in \mathbb{N}^n, \|m\|_1 \leq l} \frac{D^m(F)(x')}{m!} \lim_{l \to \infty} \mathrm{I}^k(x, x', (y - x')^m)
$$

The last line is determined by the action of $\mathrm{I}^k$ on elements of the set $\{(x, x', F) : F(y) = (y - x')^m, m \in \mathbb{N}^n\}$, concluding the proof.

□

## E.4. Proof of Corollary 3

Sundararajan et al. (2017) has shown that IG is linear and Eq. (8) shows the actions of IG on polynomials.

Let $F \in C^\omega$ and let $T_l$ be the Taylor approximation of $F$ of order $l$ centered at $x'$. It is known that $\frac{\partial T_l}{\partial x_i} \to \frac{\partial F}{\partial x_i}$ uniformly on a compact domain, such as $[a, b]$. Thus,

$$
\lim_{l \to \infty} \mathrm{IG}_i(x, T_l) = \lim_{l \to \infty} (x_i - x'_i) \int_0^1 \frac{\partial T_l}{\partial x_i}(x' + t(x - x'))dt
$$

$$
= (x_i - x'_i) \int_0^1 \frac{\partial F}{\partial x_i}(x' + t(x - x'))dt \tag{30}
$$

$$
= \mathrm{IG}_i(x, F)
$$

Thus IG satisfies the continuity criteria. Apply Theorem 2 for result.

## E.5. Integrated Hessian and Augmented Integrated Hessian

E.5.1. DEFINITION OF INTEGRATED HESSIAN

Here we give a complete definition of IH and detail how IH distributes monomials. We also detail $\mathrm{IH}^*$ and show it satisfies Corollary 4. We then show both satisfy completeness, linearity, null feature, and symmetry, and augmented IH satisfies baseline test for interactions ($k \leq n$).

Let $\sigma^k_T$ be the set of sequences of length $k$ such that the sequence is made of the elements of $T \neq \emptyset$ and each element appears at least once. For example, $\sigma^3_{\{1,2\}} = \{(1, 1, 2), (1, 2, 1), (1, 2, 2), (2, 1, 1), (2, 1, 2), (2, 2, 1)\}$. For

a given sequence $s$, define $\text{IG}_s(x, F)$ to be a recursive implementation of IG according to the sequence $s$, i.e., $\text{IG}_{(1,2,3)}(x, F) = \text{IG}_3(x, IG_2(\cdot, \text{IG}_1(\cdot, F)))$.

We can then define the $k$-order Integrated Hessian for $T \neq \emptyset$ by:

$$\text{IH}_T^k(x, F) = \sum_{s \in \sigma_T^k} \text{IG}_s(x, F), \tag{31}$$

and for $T = \emptyset$, we define $\text{IH}_\emptyset^k(x, x', F) = F(x')$.

### E.5.2. IH POLICY DISTRIBUTING MONOMIALS AND CONTINUITY CONDITION

We now move to inspect this equation and establish some properties. First, IG is linear, establishing that IH is also linear by its form.

Next, we establish its policy distributing monomials centred at $x'$. Eq. (8) states that for a monomial $F(y) = (y - x')^m$, $m \neq 0$,

$$\text{IG}_i(x, x', (y - x')^m) = \begin{cases} \frac{m_i}{\|m\|_1}(y - x')^m & \text{if } i \in S_m \\ 0 & \text{if } i \notin S_m \end{cases} \tag{32}$$

Then for a given sequence $s \in \sigma_T^k$ and synergy function $(y - x')^m$, $T \subseteq S_m$,

$$
\begin{aligned}
\text{IG}_s(x, (y - x')^m) &= \text{IG}_{s_k}(x, \text{IG}_{s_{k-1}}(...\text{IG}_{s_1}(\cdot, (y - x')^m)....) \\
&= \text{IG}_{s_k}(x, \text{IG}_{s_{k-1}}(...\text{IG}_{s_2}(\cdot, \frac{m_{s_1}(y - x')^m}{\|m\|_1})....) \\
&= \text{IG}_{s_k}(x, \text{IG}_{s_{k-1}}(...\text{IG}_{s_3}(\cdot, \frac{m_{s_1}m_{s_2}(y - x')^m}{\|m\|_1^2})....) \\
&= ... \\
&= \text{IG}_{s_k}(x, \frac{\Pi_{1 \leq i \leq k-1}m_{s_i}(y - x')^m}{\|m\|_1^{k-1}}) \\
&= \frac{\Pi_{1 \leq i \leq k}m_{s_i}}{\|m\|_1^k}(x - x')^m
\end{aligned}
\tag{33}
$$

However, if there exists any elements of $s$ that is not in $S_m$, then:

$$\text{IG}_s(x, x', (y - x')^m) = 0, \tag{34}$$

due to some $s_j \notin S_m$ in the sequence.

Now, applying the definition of IH when $m \neq 0$, we get:

$$
\begin{aligned}
\text{IH}_T^k(x, (y - x')^m) &= \sum_{s \in \sigma_T^k} \text{IG}_s(x, (y - x')^m) \\
&= \begin{cases} \sum_{s \in \sigma_T^k} \frac{\Pi_{1 \leq i \leq k}m_{s_i}}{\|m\|_1^k}(x - x')^m & \text{if } T \subseteq S_m \\ \sum_{s \in \sigma_T^k} 0 & \text{if } T \nsubseteq S_m \end{cases} \\
&= \begin{cases} \frac{M_T^k(m)}{\|m\|_1^k}(x - x')^m & \text{if } T \subseteq S_m \\ 0 & \text{if } T \nsubseteq S_m, \end{cases}
\end{aligned}
\tag{35}
$$

where we define $M_T^k(m) = \sum_{|l|=k \text{ s.t. } S_l=T} \binom{k}{l}m^l$, with $\binom{k}{l} = \frac{k!}{\Pi_{i \in S_l}l_i!}$ the multinomial coefficient. In the case $T = S_m = \emptyset$, we set $\frac{M_T^k(m)}{\|m\|_1^k} = 1$.

Now let us turn to the question of the continuity of Taylor approximation for analytic functions. Let $T_l$ be the Taylor approximation of some $F \in \mathcal{C}^\omega$. Using Corollary 3, we have $\lim_{l \to \infty} \text{IG}_i(x, T_l) = \text{IG}_i(x, F)$. This implies:

$$
\begin{aligned}
\text{IG}_i(x, F) &= \lim_{l \to \infty} \text{IG}_i(x, T_l) \\
&= \sum_{m \in \mathbb{N}^n} \frac{D^m(F)(x')}{m!} \text{IG}_i(x, (y - x')^m) \\
&= \sum_{m \in \mathbb{N}^n} \frac{D^m(F)(x')}{m!} \frac{m_i}{\|m\|_1} (x - x')^m
\end{aligned}
\tag{36}
$$

That is, the above sum is convergent for all $x \in [a, b]$, implying that $\text{IG}_i(\cdot, F) \in \mathcal{C}^\omega$. Also note:

$$
\text{IG}_i(x, T_l) = \sum_{m \in \mathbb{N}^n, |m| \leq l} \frac{D^m(F)(x')}{m!} \frac{m_i}{\|m\|_1} (x - x')^m
\tag{37}
$$

This shows that $\text{IG}(x, T_l)$ is a Taylor approximation of $\text{IG}_i(x, F)$. Thus, for $F \in \mathcal{C}^\omega$ and a sequence $s$, we can pull the limit out consecutively since we are simply dealing with a series of Taylor approximations.

$$
\begin{aligned}
\text{IG}_s(x, F) &= \text{IG}_{s_k}(x, \text{IG}_{s_{k-1}}(...\text{IG}_{s_1}(\cdot, F)...)) \\
&= \text{IG}_{s_k}(x, \text{IG}_{s_{k-1}}(... \lim_{l \to \infty} \text{IG}_{s_1}(\cdot, T_l)...)) \\
&= \text{IG}_{s_k}(x, \text{IG}_{s_{k-1}}(... \lim_{l \to \infty} \text{IG}_{s_2}(\cdot, \text{IG}_{s_1}(\cdot, T_l))...)) \\
&= \lim_{l \to \infty} \text{IG}_{s_k}(x, \text{IG}_{s_{k-1}}(...\text{IG}_{s_1}(\cdot, T_l)...)) \\
&= \lim_{l \to \infty} \text{IG}_s(x, T_l),
\end{aligned}
\tag{38}
$$

which establishes that $\text{IH}^k$ satisfies the continuity property. This implies the following corollary:

**Corollary 7.** Integrated Hessian of order $k$ is the unique $k^{\text{th}}$-order method to satisfy linearity, the continuity condition, and distributes monomials as in Eq. (35).

E.5.3. ESTABLISHING FURTHER PROPERTIES OF IH

To show IH is complete, observe for a monomial $F(y) = (y - x')^m$, $m \neq 0$,

$$
\begin{aligned}
\sum_{S \in \mathcal{P}_k, |S| > 0} \text{IH}_S^k(x, x', F) &= \sum_{S \subseteq S_m, |S| > 0} \frac{M_T^k(m)}{\|m\|_1^k} (x - x')^m \\
&= \sum_{S \subseteq S_m, |S| > 0} \frac{\sum_{|l| = k \text{ s.t. } S_l = S} \binom{k}{l} m^l}{\|m\|_1^k} (x - x')^m \\
&= \frac{\|m\|_1^k}{\|m\|_1^k} (x - x')^m \\
&= (x - x')^m
\end{aligned}
$$

When $m = 0$, we get $\text{IH}_S^k(x, x', F) = 0$ except when $S = \emptyset$, in which case we get $\text{IH}_S^k(x, x', F) = 1$.

Applying the Taylor decomposition of $F$ and continuity property to a general $F \in \mathcal{C}^\omega$, we get:

$$\sum_{S \in \mathcal{P}_k, |S| > 0} \mathrm{IH}_S^k(x, x', F) = \sum_{S \in \mathcal{P}_k, |S| > 0} \lim_{l \to \infty} \mathrm{IH}_S^k(x, x', T_l)$$

$$= \lim_{l \to \infty} \sum_{S \in \mathcal{P}_k, |S| > 0} \sum_{m \in \mathbb{N}^n, 0 < \|m\|_1 \le l} \frac{D^m(F)(x')}{m!} \mathrm{IH}_S^k(x, x', (y - x')^m)$$

$$= \lim_{l \to \infty} \sum_{m \in \mathbb{N}^n, 0 < \|m\|_1 \le l} \frac{D^m(F)(x')}{m!} \sum_{S \in \mathcal{P}_k, |S| > 0} \mathrm{IH}_S^k(x, x', (y - x')^m)$$

$$= \lim_{l \to \infty} \sum_{m \in \mathbb{N}^n, 0 < \|m\|_1 \le l} \frac{D^m(F)(x')}{m!} (x - x')^m$$

$$= \lim_{l \to \infty} \sum_{m \in \mathbb{N}^n, \|m\|_1 \le l} \frac{D^m(F)(x')}{m!} (x - x')^m - F(x')$$

$$= F(x) - F(x')$$

To show IH satisfies null feature, we proceed as in the proof for Recursive Shapley and suppose that $F$ does not vary in $x_i$. Then for any $S \in \mathcal{P}_k$ such that $i \in S$, $\phi_S(F) = 0$ since the synergy function is an interaction satisfying null feature. Then if $i \in T$,

$$\mathrm{IH}_T^k(x, F) = \sum_{S \in \mathcal{P}_k} \mathrm{IH}_T^k(x, \phi_S(F))$$

$$= \sum_{S \in \mathcal{P}_k \text{ s.t. } i \in S} \mathrm{IH}_T^k(x, \phi_S(F)) + \sum_{S \in \mathcal{P}_k \text{ s.t. } i \notin S} \mathrm{IH}_T^k(x, \phi_S(F)) \qquad (39)$$

$$= \sum_{S \in \mathcal{P}_k \text{ s.t. } i \in S} \mathrm{IH}_T^k(x, 0) + \sum_{S \in \mathcal{P}_k \text{ s.t. } i \notin S} 0$$

$$= 0$$

To show symmetry, let $\pi$ be a permutation. Note that since $(\pi y)_{\pi(i)} = y_i$, we also have $(\pi^{-1} y)_i = (\pi^{-1} y)_{\pi^{-1}(\pi(i))} = y_{\pi(i)}$. Then, if $F(y) = (y - x')^m$, we get

$$F \cdot \pi^{-1}(y) = (y_{\pi(1)} - x_1')^{m_1} \cdots (y_{\pi(n)} - x_n')^{m_n}$$

$$= (y_1 - x_{\pi^{-1}(1)}')^{m_{\pi^{-1}(1)}} \cdots (y_n - x_{\pi^{-1}(n)}')^{m_{\pi^{-1}(n)}}$$

$$= (y - \pi x')^{\pi m}$$

Also note that,

$$S_{\pi m} = \{i : (\pi m)_i > 0\}$$

$$= \{i : m_{\pi^{-1}(i)} > 0\}$$

$$= \{\pi(i) : m_{\pi^{-1}(\pi(i))} > 0\}$$

$$= \{\pi(i) : m_i > 0\}$$

$$= \{\pi(i) : i \in S_m\}$$

$$= \pi S_m$$

Then,

$$\text{IH}^k_{\pi T}(\pi x, \pi x', F \circ \pi^{-1}) = \begin{cases} \frac{M^k_{\pi T}(\pi m)}{\|\pi m\|^k_1}(\pi x - \pi x')^{\pi m} & \text{if } \pi T \subseteq S_{\pi m} \\ 0 & \text{if } \pi T \nsubseteq S_{\pi m} \end{cases}$$

$$= \begin{cases} \frac{M^k_T(m)}{\|m\|^k_1}(x - x')^m & \text{if } T \subseteq S \\ 0 & \text{if } T \nsubseteq S \end{cases}$$

$$= \text{IH}^k_T(x, x', F)$$

Now, if we take $\pi \in \mathcal{C}^\omega$ and denote $\pi^{-1}_j$ to be the $j^{\text{th}}$ output of $\pi^{-1}$, then $\frac{\partial \pi^{-1}_j}{\partial x_i} = \mathbb{1}_{j=\pi^{-1}(i)}$. Then we have

$$\frac{\partial(F \circ \pi^{-1})}{\partial x_i}(y) = \sum_{j=1}^n \frac{\partial F}{\partial x_j}(\pi^{-1}(y))\frac{\partial \pi^{-1}_j}{\partial x_i}(y)$$

$$= \frac{\partial F}{\partial x_{\pi^{-1}(i)}}(\pi^{-1}(y)),$$

which yields

$$D^{\pi m}(F \circ \pi^{-1})(\pi x') = \frac{\partial^{\|\pi m\|_1}(F \circ \pi^{-1})}{\partial x_1^{(\pi m)_1} \cdots \partial x_n^{(\pi m)_n}}(\pi x')$$

$$= \frac{\partial^{\|\pi m\|_1} F}{\partial x_{\pi^{-1}(1)}^{m_{\pi^{-1}(1)}} \cdots \partial x_{\pi^{-1}(n)}^{m_{\pi^{-1}(n)}}}(\pi^{-1}\pi x')$$

$$= \frac{\partial^{\|m\|_1} F}{\partial x_1^{m_1} \cdots \partial x_n^{m_n}}(x')$$

$$= D^m F(x')$$

From the above we have for general $F$,

$$\text{IH}^k_{\pi S}(\pi x, \pi x', F \circ \pi^{-1}) = \lim_{l \to \infty} \text{IH}^k_{\pi S}\left(\pi x, \pi x', \sum_{m \in \mathbb{N}^n, 0 < \|m\|_1 \leq l} \frac{D^m(F \circ \pi^{-1})(\pi x')}{m!}(y - \pi x')^m\right)$$

$$= \lim_{l \to \infty} \sum_{m \in \mathbb{N}^n, 0 < \|m\|_1 \leq l} \frac{D^m(F \circ \pi^{-1})(\pi x')}{m!}\text{IH}^k_{\pi S}(\pi x, \pi x', (y - \pi x')^m)$$

$$= \lim_{l \to \infty} \sum_{m \in \mathbb{N}^n, 0 < \|m\|_1 \leq l} \frac{D^{\pi m}(F \circ \pi^{-1})(\pi x')}{(\pi m)!}\text{IH}^k_{\pi S}(\pi x, \pi x', (y - \pi x')^{\pi m})$$

$$= \lim_{l \to \infty} \sum_{m \in \mathbb{N}^n, 0 < \|m\|_1 \leq l} \frac{D^m(F)(x')}{m!}\text{IH}^k_S(x, x', (y - x')^m)$$

$$= \lim_{l \to \infty} \text{IH}_S(x, x', T_l)$$

$$= \text{IH}_S(x, x', F)$$

### E.5.4. AUGMENTED INTEGRATED HESSIAN AND ITS PROPERTIES

The synergy function $\phi$ is taken implicitly with respect to a baseline appropriate to $F$. To make the baseline choice explicit, we write $\phi(F) = \phi(x', F)$. Augmented Integrated Hessian is then defined as:

$$\text{IH}^{k*}_T(x, x', F) = \phi_T(x', F)(x) + \text{IH}^k_T\left(x, x', F - \sum_{S \in \mathcal{P}_k} \phi_S(x', F)\right) \tag{40}$$

As in Augmented Recursive Shapley, Augmented Integrated Hessian explicitly distributes $\phi_T(F)$ to group $T$ when $|T| \leq k$, and distributes $\phi_T(F)$ as IH when $|T| > k$.

To establish the monomial distribution policy we inspect the action of $\text{IH}_T^{k*}$ in different cases. Plugging in $(y - x')^m$ to the above, if $|S_m| \leq k$, the right term is zero and Eq. (10) holds, while if $|S_m| > k$, the left term is zero and the right term is $\text{IH}_T^k(x, (y - x')^m)$. It is also easy to see that the above is linear.

Regarding the continuity condition, observe that:

$$\phi_S(F) = \sum_{m \in \mathbb{N}^n, S_m = S} \frac{D^m(F)(x')}{m!}(x - x')^m$$

$$= \lim_{l \to \infty} \sum_{m \in \mathbb{N}^n, \|m\|_1 \leq l, S_m = S} \frac{D^m(F)(x')}{m!}(x - x')^m$$

$$= \lim_{l \to \infty} \phi_S(T_l),$$

which gains,

$$\lim_{l \to \infty} \text{IH}_S^{k*}(x, T_l) = \lim_{l \to \infty} \phi_S(T_l)(x) + \text{IH}_S^k(x, T_l - \sum_{R \in \mathcal{P}_k} \phi_R(T_l))$$

$$= \phi_S(F)(x) + \text{IH}_S^k(x, \lim_{l \to \infty} T_l - \sum_{S \in \mathcal{P}_k} \phi_R(T_l))$$

$$= \text{IH}_S^k(x, F - \sum_{R \in \mathcal{P}_k} \phi_R(F))$$

$$= \text{IH}_S^{k*}(x, F),$$

which establishes Corollary 4.

To show completeness, consider the decomposition $F = \sum_{S \in \mathcal{P}_n} \phi_S(F)$. Now $\text{IH}^{k*}$ satisfies completeness for the subset of functions $\Phi_S \in C_S, |S| \leq k$ from the completeness of $\phi$ and Eq. (40). Also, $\text{IH}^{k*}$ satisfies completeness for the subset of functions $\Phi_S \in C_S, |S| > k$ because $\text{IH}^k$ satisfies completeness. From this we have:

$$\sum_{T \in \mathcal{P}_k, |T| \neq 0} \text{IH}_T^{k*}(x, x', F) = \sum_{T \in \mathcal{P}_k, |T| \neq 0} \text{IH}_T^{k*}(x, x', \sum_{S \in \mathcal{P}_n} \phi_S(F))$$

$$= \sum_{S \in \mathcal{P}_n} \sum_{T \in \mathcal{P}_k, |T| \neq 0} \text{IH}_T^{k*}(x, x', \phi_S(F))$$

$$= \sum_{S \in \mathcal{P}_n, |S| \neq 0} [\phi_S(F)(x) - \phi_S(F)(x')]$$

$$= \sum_{S \in \mathcal{P}_n, |S| \neq 0} [\phi_S(F)(x)] + F(x') - F(x')$$

$$= \sum_{S \in \mathcal{P}_n} [\phi_S(F)(x)] - F(x')$$

$$= F(x) - F(x')$$

Baseline test for interactions applies immediately from the definition of Augmented Integrated Hessian in Eq. (40). Concerning null feature, suppose $F$ does not vary in some $x_i$ and $i \in T$. First, we have $\phi_T(F) = 0$. Also, $F - \sum_{R \in \mathcal{P}_k} \phi_R(F)$ does not vary in $x_i$ either, so, since $\text{IH}^k$ satisfies null feature. Thus we have $\text{IH}^{k*}(x, F) = 0$ by Eq. (40).

Lastly, concerning symmetry, let $\pi$ be a permutation. Note that $\phi$ is symmetric, as it is the $k = n$ case for Shapley-Taylor, which is symmetric. Then,

$$\begin{aligned}
\text{IH}_{\pi T}^{k*}(\pi x, \pi x', F \circ \pi^{-1}) &= \phi_{\pi T}(\pi x', F \circ \pi^{-1})(\pi x) + \text{IH}_{\pi T}^k(\pi x, \pi x', F \circ \pi^{-1} - \sum_{R \in \mathcal{P}_k} \phi_{\pi R}(\pi x', F \circ \pi^{-1})) \\
&= \phi_T(x', F)(x) + \text{IH}_T^k(\pi x, \pi x', \phi_{\pi R}(\pi x', \sum_{R \subset N, |R|>k} F \circ \pi^{-1})) \\
&= \phi_T(x', F)(x) + \sum_{R \subset N, |R|>k} \text{IH}_T^k(\pi x, \pi x', \phi_{\pi R}(\pi x', F \circ \pi^{-1})) \\
&= \phi_T(x', F)(x) + \sum_{R \subset N, |R|>k} \text{IH}_T^k(x, x', \phi_R(x', F)) \\
&= \phi_T(x', F)(x) + \text{IH}_T^k(x, x', \sum_{R \subset N, |R|>k} \phi_R(x', F)) \\
&= \text{IH}_T^{k*}(x, x', F)
\end{aligned}$$

**E.6. Sum of Powers**

E.6.1. DEFINING SUM OF POWERS

To define Sum of Powers, we first turn to defining a slight alteration of the Shapley-Taylor method. Suppose we performed Shapley-Taylor on a function $F$, but we treated $F$ as a function of every variable except for $x_i$, which we held at the input value. Specifically, for a given index $i$ and coalition $S$ with $i \in S$, we perform the $(|S|-1)$-order Shapley-Taylor method for the coalition $S \setminus \{i\}$. We perform this on an alteration of $F$, so that $F$ is a function of $n-1$ variables because the $x_i$ value is fixed. We denote this function $\text{ST}_S^{-i}$, which has formula:

$$\text{ST}_S^{-i}(x, x', F) = \frac{|S|-1}{n-1} \sum_{T \subseteq N \setminus S} \frac{\delta_{S \setminus \{i\}|T \cup \{i\}} F(x)}{\binom{n-2}{|T|}} \tag{41}$$

With this, we define Sum of Powers for $k \geq 2$ as:

$$\text{SP}_S^k(x, x', F) = \begin{cases} \sum_{i \in S} \left[ \text{ST}_S^{-i}(x, x', \text{IG}_i(\cdot, x', F)) \right] & \text{if } |S| = k \\ \phi_S(F) & \text{if } |S| < k \end{cases} \tag{42}$$

We define the Sum of Powers for $k = 1$ as the IG, with the addition that $\text{SP}_\emptyset^1(x, x', F) = F(x')$.

Similar to the alteration of the Shapley-Taylor, we can alter the Shapley method, giving us:

$$\text{Shap}_j^{-i}(x, x', F) = \sum_{S \subset N \setminus \{i,j\}} \frac{|S|!(n-|S|-2)!}{(n-1)!} \left( F(x_{S \cup \{i,j\}}) - F(x_{S \cup \{i\}}) \right) \tag{43}$$

For the Sum of Powers $k = 2$ case, the altered Shapley-Taylor is a 1-order Shapley-Taylor method, and conforms to the Shapley method:

$$\text{SP}_{i,j}^2(x, x', F) = \begin{cases} \text{Shap}_j^{-i}(x, x', \text{IG}_i(\cdot, x', F)) + \text{Shap}_i^{-j}(x, x', \text{IG}_j(\cdot, x', F)) & \text{if } |S| = 2 \\ \phi_S(F) & \text{if } |S| \leq 1 \end{cases} \tag{44}$$

E.6.2. PROOF OF COROLLARY 5

For the $k = 1$ case, Sum of Powers is the IG, which satisfies linearity, distributes as in Eq. 11, and satisfies the continuity condition.

We now assume $k \geq 2$ for the rest of the section. First, $\text{SP}_S^k$ satisfies linearity because IG is linear in $F$ and $\text{ST}_S^{-i}$ is linear in $F$.

We now proceed by cases to establish how $\text{SP}^k$ distributes monomials. We consider first the action of $\text{ST}_S^{-i}$ on $F(y) = (y - x')^m$. $\text{ST}_S^{-i}$ acts as the $(|S| - 1)$-order Shapley-Taylor on an augmented function $F^{-i}(y_1, ..., y_{i-1}, y_{i+1}, ..., y_i) := (x_i - x_i')^{m_i} \Pi_{j \neq i}(y_j - x_j')^{m_j}$. Now, $\Pi_{j \neq i}(y_j - x_j')^{m_j}$ is a synergy function of $S_m \setminus \{i\}$. Thus we can use the distribution rule of Shapley-Taylor, gaining

$$
\text{ST}_S^{-i}(x, x', F) = \text{ST}_{S \setminus \{i\}}^{|S|-1}(x_{-i}, x'_{-i}, F^{-i})
$$
$$
= \begin{cases} (x_i - x_i')^m & \text{if } S = S_m \\ \frac{(x_i - x_i')^m}{\binom{|S|-1}{k-1}} & \text{if } S \subsetneq S_m, |S| = k \ , \\ 0 & \text{else} \end{cases} \tag{45}
$$

where $x_{-i}$ denotes the vector $x$ with the $i^{\text{th}}$ component removed.

With this established, we now show the action of the Sum of Powers method for an exhaustive set of cases:

1. $(|S| < k, S = S_m)$: $\text{SP}_S^k(x, (y - x')^m) = \phi_S((y - x')^m) = (y - x')^m$.

2. $(|S| < k, S \neq S_m)$: $\text{SP}_S^k(x, (y - x')^m) = \phi_S((y - x')^m) = 0$.

3. $(|S| = k, S \subseteq S_m)$:

$$
\text{SP}_S^k(x, x', (y - x')^m) = \sum_{i \in S} \left[ \text{ST}_S^{-i}(x, x', \text{IG}_i(\cdot, x', (y - x')^m)) \right]
$$
$$
= \sum_{i \in S} \left[ \text{ST}_S^{-i}\left(x, x', \frac{m_i}{\|m\|_1}(y - x')^m\right) \right]
$$
$$
= \sum_{i \in S} \frac{1}{\binom{|S_m|-1}{|S|-1}} \frac{m_i}{\|m\|_1}(x - x')^m
$$
$$
= \frac{1}{\binom{|S_m|-1}{|S|-1}} \frac{\sum_{i \in S} m_i}{\|m\|_1}(x - x')^m
$$

4. $(|S| = k, S \nsubseteq S_m)$: Let $i \in S$. If $i \in S \setminus S_m$, then $\text{ST}_S^{-i}(x, x', \text{IG}_i(\cdot, x', (y - x')^m)) = \text{ST}_S^{-i}(x, x', 0)) = 0$.

   If, on the other hand, $i \in S_m$, then $\text{ST}_S^{-i}(x, x', \text{IG}_i(\cdot, x', (y - x')^m)) = \text{ST}_S^{-i}(x, x', \frac{m_i}{\|m\|_1}(y - x')^m)$. Now, the altered Shapley-Taylor takes the value of zero for synergy functions of sets that are not super-sets of the attributed group, $S \setminus \{i\}$. Also, $(y - x')^m$ is a synergy function of $S_m$, and $S_m$ is not a super-set of $S \setminus \{i\}$. Thus $\text{ST}_S^{-i}(x, x', \frac{m_i}{\|m\|_1}(y - x')^m) = 0$.

   This established that each term in the sum $\sum_{i \in S} \left[ \text{ST}_S^{-i}(x, x', \text{IG}_i(\cdot, x', (y - x')^m)) \right]$ is zero, gaining $\text{SP}_S^k(x, x', (y - x')^m) = 0$.

Thus Sum of Powers has a distribution scheme that agrees with Eq. (11). To restate:

$$
\text{SP}_T^k(x, (y - x)^m) = \begin{cases} (x - x')^m & \text{if } T = S_m \\ \frac{1}{\binom{|S_m|-1}{k-1}} \frac{\sum_{i \in T} m_i}{\|m\|_1}(x - x')^m & \text{if } T \subsetneq S_m, |T| = k \\ 0 & \text{else} \end{cases} \tag{46}
$$

Finally, IG satisfies the continuity condition by Corollary 3, and it is easy to see that that $\text{ST}_S^{-1}$ satisfies the continuity condition. Thus Sum of Powers obeys the continuity condition.

1485 E.6.3. ESTABLISHING FURTHER PROPERTIES FOR SUM OF POWERS

1487 To establish null feature, let $F$ not vary in $x_i$ and let $i \in S$. Sum of Powers satisfies the continuity condition, so

$$\mathrm{SP}_S^k(x, x', F) = \lim_{l \to \infty} \sum_{m \in \mathbb{N}^n, |m| \le l} \frac{D^m F(x')}{m!} \mathrm{SP}_S^k(x, x', (y - x')^m)$$

$$= \lim_{l \to \infty} \sum_{m \in \mathbb{N}^n, |m| \le l, m_i = 0} \frac{D^m F(x')}{m!} \mathrm{SP}_S^k(x, x', (y - x')^m)$$

$$= 0,$$

1498 where the second line is because $D^m F(x') = 0$ if $m_i > 0$ because $F$ does not vary in $x_i$, and the third line is because
1499 $\mathrm{SP}_S^k(x, x', (y - x')^m) = 0$ if $m_i = 0$.

1501 To establish baseline test for interaction ($k \le n$), let $\Phi_S \in \mathcal{C}^\omega$ be a synergy function of $S$ and let $T \subsetneq S$, $|T| < k$. Then
1502 $\mathrm{SP}_T^k(x, \Phi_S) = \phi_T(\Phi_S)(x) = 0$.

1503 To establish completeness, consider $F(y) = (y - x')^m$, with $|S_m| > k$. Then,

$$\sum_{S \in \mathcal{P}_k, |S| > 0} \mathrm{SP}_S^k(x, x', F) = \sum_{S \subsetneq S_m, |S| = k} \mathrm{SP}_S^k(x, x', (y - x')^m)$$

$$= \sum_{S \subsetneq S_m, |S| = k} \frac{1}{\binom{|S_m| - 1}{k - 1}} \frac{\sum_{i \in S} m_i}{\|m\|_1} (x - x')^m$$

$$= \frac{(x - x')^m}{\binom{|S_m| - 1}{k - 1} \|m\|_1} \sum_{S \subsetneq S_m, |S| = k} \sum_{i \in S} m_i$$

$$= \frac{(x - x')^m}{\binom{|S_m| - 1}{k - 1} \|m\|_1} \binom{|S_m| - 1}{k - 1} \|m\|_1$$

$$= F(x) - F(x')$$

1519 Now treating a general $F \in \mathcal{C}^\omega$, the proof is identical to the proof for Integrated Hessian,

$$\sum_{S \in \mathcal{P}_k, |S| > 0} \mathrm{SP}_T^k(x, x', F) = \sum_{S \in \mathcal{P}_k, |S| > 0} \lim_{l \to \infty} \mathrm{SP}_T^k(x, x', T_l)$$

$$= \lim_{l \to \infty} \sum_{S \in \mathcal{P}_k, |S| > 0} \sum_{m \in \mathbb{N}^n, 0 < \|m\|_1 \le l} \frac{D^m(F)(x')}{m!} \mathrm{SP}_T^k(x, x', (y - x')^m)$$

$$= \lim_{l \to \infty} \sum_{m \in \mathbb{N}^n, 0 < \|m\|_1 \le l} \frac{D^m(F)(x')}{m!} \sum_{S \in \mathcal{P}_k, |S| > 0} \mathrm{SP}_T^k(x, x', (y - x')^m)$$

$$= \lim_{l \to \infty} \sum_{m \in \mathbb{N}^n, 0 < \|m\|_1 \le l} \frac{D^m(F)(x')}{m!} (x - x')^m$$

$$= \lim_{l \to \infty} \sum_{m \in \mathbb{N}^n, \|m\|_1 \le l} \frac{D^m(F)(x')}{m!} (x - x')^m - F(x')$$

$$= F(x) - F(x')$$

1537 To show symmetry, the proof parallels the proof for Integrated Hessian in section E.5.3. Let $\pi$ be a permutation. If we let
1538 $F(y) = \overline{(y - x')^m}$ and follow what was previously established in section E.5.3, then

$$\mathrm{SP}_{\pi T}^k(\pi x, \pi x', F \circ \pi^{-1}) = \begin{cases} (\pi x - \pi x')^{\pi m} & \text{if } \pi T = S_{\pi m} \\ \frac{1}{\binom{|S_{\pi m}|-1}{k-1}} \frac{\sum_{i \in \pi T}(\pi m)_i}{\|\pi m\|_1}(\pi x - \pi x')^{\pi m} & \text{if } \pi T \subsetneq S_{\pi m}, |\pi T| = k \\ 0 & \text{else} \end{cases}$$

$$= \begin{cases} (x - x')^m & \text{if } T = S_m \\ \frac{1}{\binom{|S_m|-1}{k-1}} \frac{\sum_{i \in T} m_i}{\|m\|_1}(x - x')^m & \text{if } T \subsetneq S_m, |T| = k \\ 0 & \text{else} \end{cases}$$

$$= \mathrm{SP}_T^k(x, x', F)$$

From the above we have for general $F$,

$$\mathrm{SP}_{\pi S}^k(\pi x, \pi x', F \circ \pi^{-1}) = \lim_{l \to \infty} \mathrm{SP}_{\pi S}^k\left(\pi x, \pi x', \sum_{m \in \mathbb{N}^n, 0 < \|m\|_1 \leq l} \frac{D^m(F \circ \pi^{-1})(\pi x')}{m!}(y - \pi x')^m\right)$$

$$= \lim_{l \to \infty} \sum_{m \in \mathbb{N}^n, 0 < \|m\|_1 \leq l} \frac{D^m(F \circ \pi^{-1})(\pi x')}{m!} \mathrm{SP}_{\pi S}^k(\pi x, \pi x', (y - \pi x')^m)$$

$$= \lim_{l \to \infty} \sum_{m \in \mathbb{N}^n, 0 < \|m\|_1 \leq l} \frac{D^{\pi m}(F \circ \pi^{-1})(\pi x')}{(\pi m)!} \mathrm{SP}_{\pi S}^k(\pi x, \pi x', (y - \pi x')^{\pi m})$$

$$= \lim_{l \to \infty} \sum_{m \in \mathbb{N}^n, 0 < \|m\|_1 \leq l} \frac{D^m(F)(x')}{m!} \mathrm{SP}_S^k(x, x', (y - x')^m)$$

$$= \lim_{l \to \infty} \mathrm{SP}(x, x', T_l)$$

$$= \mathrm{SP}(x, x', F)$$

# F. Experimental Details and Additional Results

All experiments are conducted on a device with a 6-core Intel Core i7-8700.

## F.1. Model Description and Experimental Details

### F.1.1. 2-LAYER PERCEPTRON

We use a 2-layer perceptron with 64 neurons in the first layer and 32 neurons in the second layer. For activation, we use SoftPlus

$$\mathrm{SoftPlus}(x) = \frac{1}{\beta} \log\left(1 + \exp\left(\beta x\right)\right)$$

with $\beta = 5$ after each layer. We optimize using the Adam algorithm with the default hyper-parameters (Kingma & Ba, 2014) and the learning rate of 0.1054. We train the model for 1000 epochs with the whole training data, and the network achieves a test Mean-Absolute-Error (MAE) of 3.10 and a test Root-Mean-Squared-Error (MRSE) of 4.14.

**Hyperparameter tuning**: The number of neurons in each layer includes values $8, 16, 32, 64,$ and $128$ such that the size of the first hidden layer should be larger than or equal to the size of the second layer. For each dimension of the neural network, we swept through a range of stepsizes and values of $\beta$ to find the (approximately) optimal stepsize and $\beta$. The stepsize grid consists of 5 evenly spaced points between $e^{-6}$ and $e^{-1}$. The $\beta$ parameter of the SoftPlus activation includes values of 1 and 5.

### F.1.2. SECOND-DEGREE POLYNOMIAL REGRESSION

We use the *LinearRegression* function from scikit-learn (Pedregosa et al., 2011) with default values to train the polynomial regression model.

### F.2. Description of the Dataset

The Physicochemical Properties of Protein Tertiary Structure data is available at `https://archive.ics.uci.edu/ml/datasets/Physicochemical+Properties+of+Protein+Tertiary+Structure`. After preprocessing, there were a total of 9 input features from this dataset and it contained around 45,730 entries in total. The regression task is to predict the size of the residue. The list of features:

1. Total surface area (mean: 9871.60$\pm$ standard deviation: 4058.14)

2. Non polar exposed area ($3017.37 \pm 1464.32$)

3. Fractional area of exposed non polar residue ($0.30 \pm 0.06$)

4. Fractional area of exposed non polar part of residue ($103.49 \pm 55.42$)

5. Molecular mass weighted exposed area ($1.37\mathrm{e}{+}06 \pm 5.64\mathrm{e}{+}05$)

6. Average deviation from standard exposed area of residue ($145.64 \pm 70.00$)

7. Euclidian distance ($3989.76 \pm 1993.57$)

8. Secondary structure penalty ($69.98 \pm 56.49$)

9. Spacial Distribution constraints (N, K Value) ($34.52 \pm 5.98$)

**Preprocessing**: We standardize the numerical data to have mean zero and unit variance. We utilize a 70/15/15 train/validation/test split for data.

### F.3. More Details on Generating Attribution and Interaction Values

To generate the attributions using Integrated Gradient and compute the interactions utilizing Integrated Hessian and Sum of Powers, we use 200 samples from the dataset. We use numerical integration with 500 samples to approximate the integral in Integrated Gradient and Integrated Hession.

### F.4. Standard Deviation of the Interaction Values

Figure 5 demonstrates the standard deviation of the interaction values from Integrated Hessian and Sum of Powers. We notice that the standard deviation of feature 1 and feature 6 is much higher in Sum of Powers than in Integrated Hessian. Furthermore, we see that small mean interaction values (see Figure 1 and Figure 2) do not imply low interaction between features, as they can have large standard deviation values (e.g., feature 1 and feature 4).

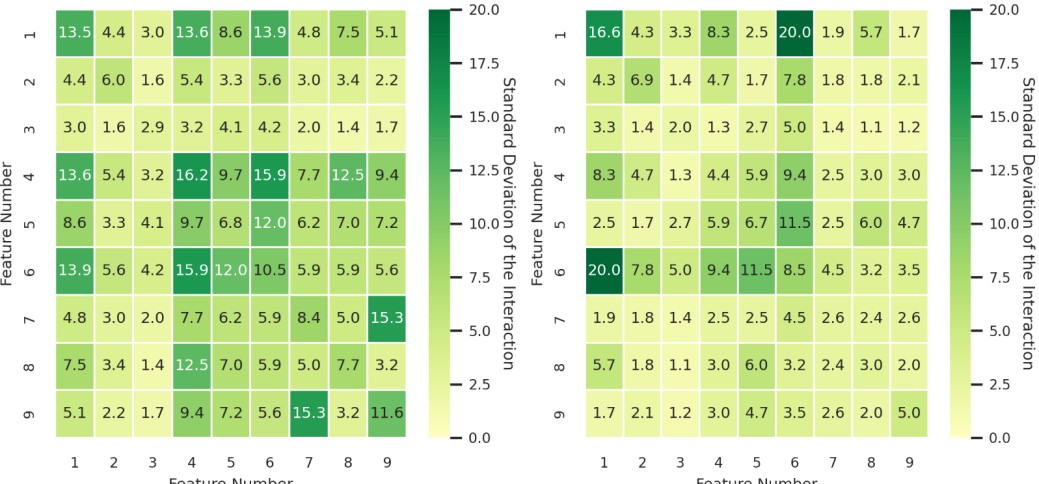

*Figure 5.* Standard deviation of interaction values. Left: Integrated Hessian. Right: Sum of Powers.

### F.5. Attribution Values

The attribution values of each feature based on Integrated Gradient are displayed in Figure 6. The features are ordered by their importance in predicting the target. The attribution values indicate the direction and magnitude of the feature's influence on the size of the residue (positive values imply an increase, negative values imply a decrease). The positive trend observed for total surface area suggests that a larger total surface area is associated with a larger size of the residue, which is consistent with intuition.

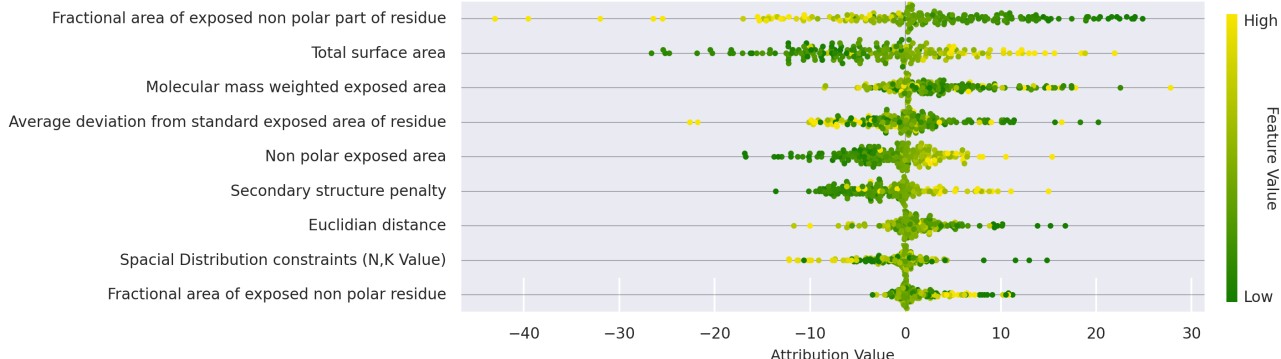

*Figure 6.* Attributions by Integrated Hessian.

