# OpenReview forum: "A Unifying Framework to the Analysis of Interaction Methods using Synergy Functions"
_ICML.cc/2023/Workshop/IMLH — IMLH 2023 Oral_

### Official Review · Reviewer_gwXL · 2023-06-15
**This paper proposes a unifying framework to characterize attribution and interaction of a continuous-input models through the concept of synergy functions.**

**Rating:** 9
**Confidence:** 4

**Review:**

This is a well-written and theory-heavy paper. It is elegant and important to characterize attribution and interaction of a continuous-input models using a single concept, synergy functions in this case. By using synergy functions, the weaknesses in existing methods such as the Integrated Hessian and indicate improvements can be shown. Moreover, synergy functions also lead to new methods such as the Sum of Powers method. This work is original and significant and I just have a few suggestions.
1. Some writings look a bit redundant to me, e.g., equation (1) and equation (5) is the same. Therefore, it could save some space if the authors can be more succinct.
2. I am confused by the messages the authors try to deliver on section 6 (Empirical Evaluation). What are the main conclusions from the experiment?

---

### Official Review · Reviewer_fpJ7 · 2023-06-17
**[full] The proposed framework uniformly handles various attribution-based interpretation techniques for machine learning**

**Rating:** 7
**Confidence:** 3

**Review:**

Many methods have been developed to comprehend black-box machine learning models by quantifying attributions and interactions between inputs and outputs. The paper proposed a framework that provides a unified approach to handling these techniques.

Pros: The paper proposed a framework that provides a unified approach to handling various attribution-based interpretation techniques for machine learning.

Cons: Despite the framework's ability to provide a unified understanding, the question of determining the superior method among these techniques remains unresolved.

---

### Official Review · Reviewer_VckU · 2023-06-17
**Strong paper levergaing game theoretic methods to extend explainability methods.**

**Rating:** 9
**Confidence:** 3

**Review:**

The authors present a exposition of attribution and interaction methods to explain deep learning models. They propose synergy functions to as a means of analysis for attribution to and k th order interaction methods of continuous-input models. The methods are built upon a strong theoretical foundation, and they show various desirable properties of the method though both theory and experiment. They illustrate their method on Protein Tertiary Structure data. Overall, the details and the rigor of the writing is impressive, and it introduces novel view points to look at model explainability.

---

### Meta-Review · Area_Chair_Tdst · 2023-06-20

**Recommendation:** Accept (Oral)
**Confidence:** 3

**Metareview:**

The work presents a novel framework of synergy functions to analyze the important problem of feature interaction in XAI. The theoretical work is solid with no major issues identified. Authors are encouraged to incorporate the reviewers' comments into their revision.

---

### Decision · Program_Chairs · 2023-06-20

Accept (Oral)